# Timapiprant, a prostaglandin D2 receptor antagonist, ameliorates pathology in a rat Alzheimer's model

Charles H Wallace[1], Giovanni Oliveros[1], Peter A Serrano[2], Patricia Rockwell[1,3], Lei Xie[4,5], Maria Figueiredo-Pereira[1,3]

We investigated the relevance of the prostaglandin D2 pathway in Alzheimer's disease, because prostaglandin D2 is a major prostaglandin in the brain. Thus, its contribution to Alzheimer's disease merits attention, given the known impact of the prostaglandin E2 pathway in Alzheimer's disease. We used the TgF344-AD transgenic rat model because it exhibits age-dependent and progressive Alzheimer's disease pathology. Prostaglandin D2 levels in hippocampi of TgF344-AD and wild-type littermates were significantly higher than prostaglandin E2. Prostaglandin D2 signals through DP1 and DP2 receptors. Microglial DP1 receptors were more abundant and neuronal DP2 receptors were fewer in TgF344-AD than in wild-type rats. Expression of the major brain prostaglandin D2 synthase (lipocalin-type PGDS) was the highest among 33 genes involved in the prostaglandin D2 and prostaglandin E2 pathways. We treated a subset of rats (wild-type and TgF344-AD males) with timapiprant, a potent highly selective DP2 antagonist in development for allergic inflammation treatment. Timapiprant significantly mitigated Alzheimer's disease pathology and cognitive deficits in TgF344-AD males. Thus, selective DP2 antagonists have potential as therapeutics to treat Alzheimer's disease.

## Introduction

Alzheimer's disease (AD) is the most common type of dementia, is highly prevalent in the ageing population, and will become more prevalent as life expectancy continues to rise. AD is a multifactorial disease, and chronic neuroinflammation is recognized as a critical factor in its pathogenesis (Bronzuoli et al, 2016). A major player in inflammation is the cyclooxygenase (COX)-mediated signaling pathway, which is the principal mediator of CNS neuroinflammation (Liang et al, 2007; Cudaback et al, 2014). The COX pathway generates prostaglandins (PGs), which are bioactive signaling lipids responsible for many processes including inflammation (Bartels & Leenders, 2010). PG signaling is implicated in AD, as some PGs aggravate its pathology, whereas others may remediate it (Biringer, 2019). Based on data from epidemiological studies, there is a decreased risk of AD in patients taking NSAIDs, which are inhibitors of the COX pathway (Vlad et al, 2008). Inhibiting COXs with NSAIDs could be a promising therapeutic strategy. However, whereas long-term use of NSAIDs is associated with a reduced incidence of AD in epidemiologic studies, randomized controlled trials did not replicate these findings (Deardorff & Grossberg, 2017). Moreover, NSAIDs target COX-1 and/or COX-2 enzymes stopping most PG synthesis. Nonspecific inhibition of PG synthesis can have a variety of negative side effects because PGs have many functions including inflammation, nociception, sleep, cardiovascular maintenance, and reproduction (Narumiya et al, 1999). Accordingly, negative side effects such as renal failure, heart problems, and stroke were reported for NSAIDs during several clinical trials (Deardorff & Grossberg, 2017). Thus, NSAIDs are not recommended for either primary prevention or treatment of AD. Based on these concerns, it is important to find new targets further downstream in the COX-signaling pathway, such as specific PG signaling that can be explored for potential therapeutic intervention.

In the present study, we focused on the prostaglandin D2 (PGD2) signaling pathway because PGD2 is the most abundant prostaglandin in the brain and is the one that increases the most under neuropathological conditions (Abdel-Halim et al, 1977; Liang et al, 2005). In the brain, lipocalin-type prostaglandin D synthase (L-PGDS) is the primary synthase for PGD2 (Urade, 2021). PGD2 undergoes a nonenzymatic dehydration producing PGJ2 (Figueiredo-Pereira et al, 2014). PGD2 signals through its two antagonistic receptors, DP1 and DP2, the latter also known as CRTH2 or GPR44. DP1 receptor activation by PGD2 is coupled to the G protein $G_s$, leading to an increase in cAMP with calcium-flux (Milatovic et al, 2011; Woodward et al, 2011). DP1 plays a well characterized role in sleep function (Ahmad et al, 2019), and in vivo studies show that DP1 modulation is protective in ischemic and hemorrhagic models of stroke (Ahmad et al, 2010, 2017, 2019; Doré & Shafique Ahmad, 2015). DP2 receptor activation by PGD2 and PGJ2 is coupled to the G protein $G_i$ leading to a decrease in cAMP and an increase in calcium mobilization, both of which can lead to neuronal damage (Woodward et al, 2011). For example, in vitro studies show adverse outcomes when treating

[1]PhD Program in Biochemistry, The Graduate Center, CUNY, New York, NY, USA  [2]Department of Psychology, Hunter College, New York, NY, USA  [3]Department of Biological Sciences, Hunter College, New York, NY, USA  [4]Department of Computer Science, Hunter College, New York, NY, USA  [5]Helen and Robert Appel Alzheimer's Disease Research Institute, Feil Family Brain and Mind Research Institute, Weill Cornell Medicine, Cornell University, New York, NY, USA

Correspondence: pereira@genectr.hunter.cuny.edu

hippocampal neuronal cultures and organotypic slices with DP2 agonists (Liang et al, 2005, 2011).

The PGD2 pathway is thoroughly studied in diseases with airway inflammation and reproduction (Rossitto et al, 2015; Marone et al, 2019), but its role in AD pathology remains unclear. Investigating the relevance of the PGD2 signaling pathway in AD is important as it could lead to new therapeutic strategies to treat neuroinflammation in pre or early stages of AD, and slow down AD pathology. Because the literature on PGD2 and its relevance to AD is limited, we investigated the importance of the PGD2 pathway in AD with the TgF344-AD (Tg-AD) rat model that closely mirrors AD in humans, specifically were neuronal loss and gliosis are detected. Tg-AD rats express the Swedish mutation (KM670/671NL) of human amyloid precursor protein (APPswe), and the Δ exon 9 mutation of human presenilin-1 (PS1ΔE9), both driven by the prion promoter (Cohen et al, 2013). Tg-AD rats develop AD pathology including cerebral amyloidosis, tauopathy, gliosis, and neuronal loss, as well as cognitive deficits, all in a progressive age-dependent manner.

To investigate whether targeting the PGD2 pathway has therapeutic potential for AD, we treated a subset of rats (WT and Tg-AD males) with timapiprant (also known as OC000459), a potent and highly selective oral DP2 antagonist. Timapiprant is an indole-acetic acid derivative that potently displaces [$^3$H]PGD2 from human recombinant DP2 (Ki = 0.013 $\mu$M), rat recombinant DP2 (Ki = 0.003 $\mu$M), and human native DP2 (Th2 cell membranes [Ki = 0.004 $\mu$M]) (Pettipher et al, 2012). Moreover, timapiprant does not interfere with the ligand binding properties or functional activities of other prostanoid receptors (EP1-4 receptors, DP1, thromboxane receptor, prostacyclin receptor, and prostaglandin F receptor) (Pettipher et al, 2012). Timapiprant, which seems to be safe and well tolerated, is under development for oral treatment of patients with allergic inflammation in diseases such as asthma and allergic rhinitis (Marone et al, 2019). Many DP2 antagonists attenuate the inflammatory response in animal studies for these diseases. Some demonstrate efficacy in phase II studies in adults with asthma, and several phase III trials are evaluating the long-term safety and efficacy of these drugs in adult and pediatric patients with moderate-to-severe asthma (Marone et al, 2019).

In summary, our studies compared PGD2, PGE2, PGJ2, and thromboxane B2 concentrations; the cellular distribution of the DP1 and DP2 receptors; and mRNA profiles for 33 genes involved in the PGD2 and PGE2 pathways in the hippocampus of 11-mo-old WT versus Tg-AD rats. We compared these results with Aβ plaque burden, neuronal loss, microgliosis, and their cognitive performance. As far as we know, our studies are the first to investigate changes in the PGD2 pathway in a rat model of AD, to determine the relevance of this pathway in AD. We established that PGD2 levels in the hippocampus are at least 14.5-fold higher than those for PGE2, independently of genotype. In addition, our data revealed significant differences in DP1 and DP2 receptor levels, respectively, in microglia and neurons of Tg-AD rats compared with controls. Our transcriptome assessment identified L-PGDS as the most abundant mRNA of the 33 genes analyzed. Notably, we established that the DP2 antagonist timapiprant ameliorated the AD pathology developed by Tg-AD male rats. Overall, our studies provide novel insights for the development of therapeutics that target the PGD2 signaling pathway to treat neuroinflammation in AD.

# Results

## PGD2 is the most abundant PG in the hippocampus of WT and Tg-AD rats

Rat hippocampal tissue from 11-mo WT (n = 31) and Tg-AD (n = 32) rats was analyzed to determine PGD2, PGE2, PGJ2, and thromboxane B2 (TxB2) concentrations. Quantitative levels of the four prostanoids ranged from 0.7 to 110.2 pg/mg wet tissue. The levels measured in the order of abundance were PGD2, TxB2, PGE2, and PGJ2, at 49.1 ± 4.1, 17.0 ± 1.5, 3.4 ± 0.4, and 2.0 ± 0.2 pg/mg wet tissue for WT rats (Fig 1). Prostanoid levels were similar in Tg-AD and WT littermates. Quantitative amounts of the four prostanoids in Tg-AD rats measured in the order of abundance were PGD2, TxB2, PGE2, and PGJ2, at 43.2 ± 4.7, 14.6 ± 1.5, 2.4 ± 0.3, and 1.4 ± 0.1 pg/mg wet tissue (Fig 1). There are no significant differences in prostanoid levels between control and Tg-AD rats, except for PGJ2 levels. The latter were lower in Tg-AD than in WT rats (t = 2.668, P = 0.005), and in seven Tg-AD rats PGJ2 was not detectable. PGJ2 is produced from PGD2 by non-enzymatic dehydration and its formation in vivo remains controversial (Bell-Parikh et al, 2003; Coutinho et al, 2017). All of these values are in accordance with those previously reported for Sprague–Dawley male rat brain cortical tissue at postnatal day 16–18, measured by quantitative UPLC–MS/MS (Shaik, 2013; Shaik et al, 2014). Under normal conditions, quantitative amounts of PGD2, PGJ2, and PGE2, measured in the order of abundance were 123.7, 12.3, and 4.5 pg/mg wet tissue or 351, 36.9, and 12.8 pmol/g wet tissue (Shaik, 2013; Shaik et al, 2014). The differences between the latter study and ours can be accounted for by the prostaglandin levels being quantified in different rat strains, at different ages and in different brain regions.

Of the four prostanoids measured, PGD2 was by far the most abundant in the hippocampal tissue, as reflected in the pie graphs shown in Fig 1. These graphs represent the proportion of each of the four prostanoids relative to their total sum. For example, it is clear that PGD2 levels are 14.5-fold and 17.5-fold higher than PGE2 in WT and Tg-AD rats, respectively (Fig 1). PGD2 levels represent 68.8% and 70.1% relative to total, whereas PGE2 levels represent 4.7% and 4.0% relative to total in WT and Tg-AD rats, respectively.

The specific chromatographic profiles of calibration standards for each prostanoid is depicted in Fig 1, showing that the four prostanoids derived from arachidonic acid can be quantified reliably in rat hippocampal tissue using LC–MS/MS analysis. Under our experimental conditions, the elution sequence was identified as TxB2, PGE2, PGD2, and PGJ2.

## Tg-AD rats have enhanced microglia and DP1/microglia co-localization levels in the hippocampus

We assessed DP1 and microglia levels in the hippocampus of WT and Tg-AD rats at 11 mo of age (Fig 2: DP1, red; microglia, green; DP1/microglia co-localization, yellowish [indicated by single white arrows]). It is clear that DP1 is detected in the four discrete hippocampal regions (SB, CA1, CA3, and DG) in WT and Tg-AD rats (Fig 2). For DP1 levels, there were no significant differences between WT and Tg-AD rats, considering the four hippocampal regions individually (Fig 3, left graph for DG only and Table S1 for all). A different situation was observed for microglia, as Tg-AD rats had significantly more microglia than WT rats, in all hippocampal regions except for CA3 (Table S2). For example, Tg-AD rats had

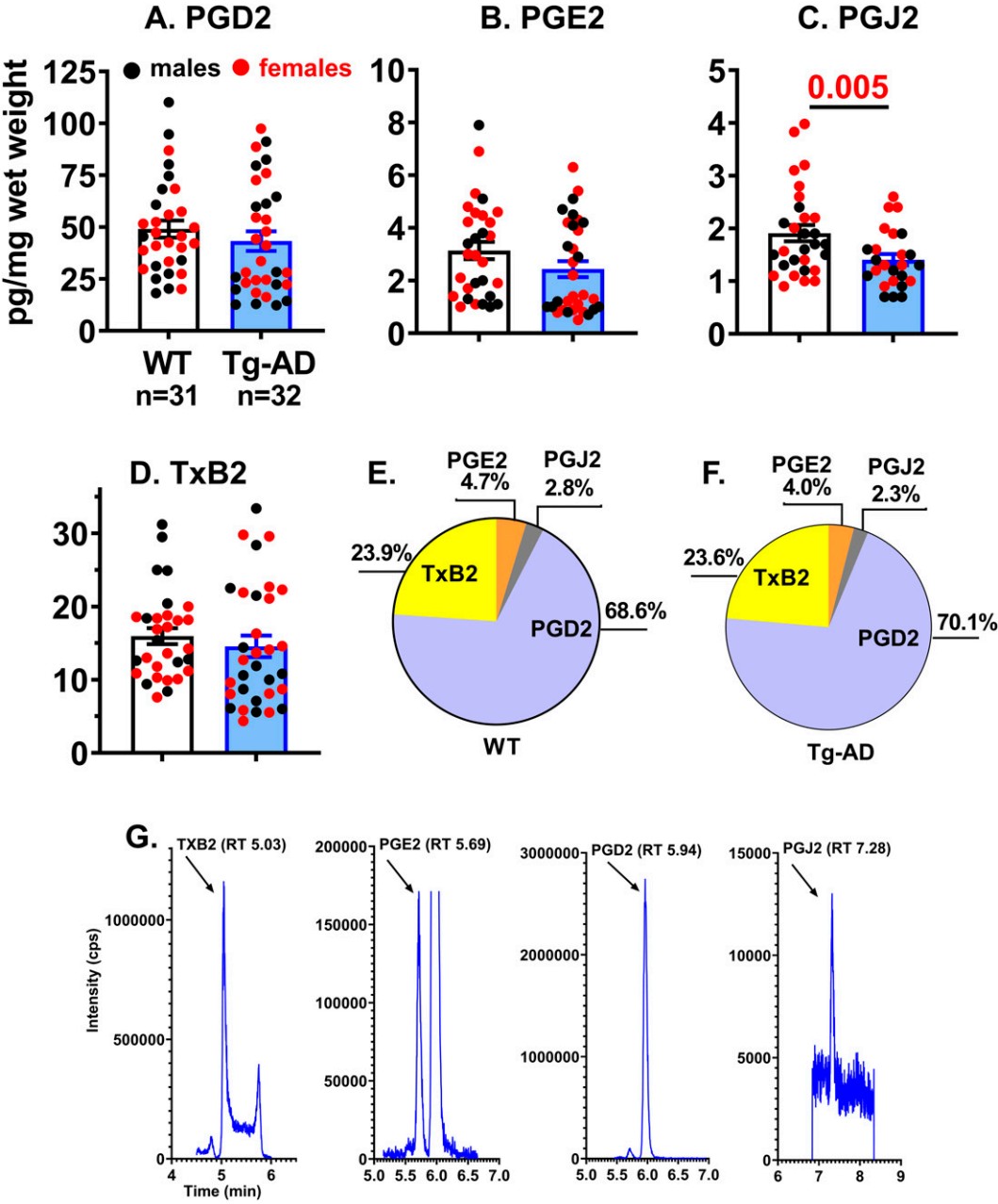

**Figure 1. PGD2 is the most abundant PG in the hippocampus of WT and Tg-AD rats.**
**(A, B, C, D)** Concentrations of prostanoids (A) PGD2, (B) PGE2, (C) PGJ2, and (D) TxB2, measured by LC–MS/MS in whole left hippocampal tissue (combined ventral and dorsal) from 11-mo WT (n = 31) and Tg-AD (n = 32) rats. Prostanoid levels were equivalent in Tg-AD and WT littermates, except for PGJ2 that were less ($t$ = 2.668, $P$ = 0.005). Significance estimated with a two-tailed Welch's $t$ test. **(E, F)** Pie graphs represent the proportion of each of the four prostanoids relative to their total sum, in (E) WT and (F) Tg-AD rats. PGD2 is the most abundant prostanoid in both WT and Tg-AD rats. **(G)** Chromatographic profiles depicting the separation of the four prostanoids using LC–MS/MS as explained under materials and methods.

more microglia (1.4-fold, $t$ = 3.15, $P$ = 0.003) in the DG hilar (HL) subregion than their WT littermates (Fig 3, middle graph for DG [HL] only).

It is evident that DP1 is co-localized with microglia in all four hippocampal regions (Fig 2), shown at higher magnification for the DG (HL) (Fig 2, bottom panels, indicated by single white arrows). Tg-AD rats had significantly higher levels (1.5-fold, $t$ = 2.99, $P$ = 0.005) of DP1/microglia co-localization than their WT littermates, only in the DG (HL) region (Fig 3, right graph for DG [HL] only, and Table S3 for all).

Microglia have a remarkable variety of morphologies associated with their specific functions, and can be divided into three phenotypes according to their cell body circularity: ramified, reactive and amoeboid (Karperien et al, 2013) (Fig 3). Most of the microglia are *ramified* with long slender processes and play a role in surveillance. *Reactive* microglia, present in intermediate numbers, exhibit shorter processes and a larger soma than ramified microglia and are in an activated state producing immune modulators.

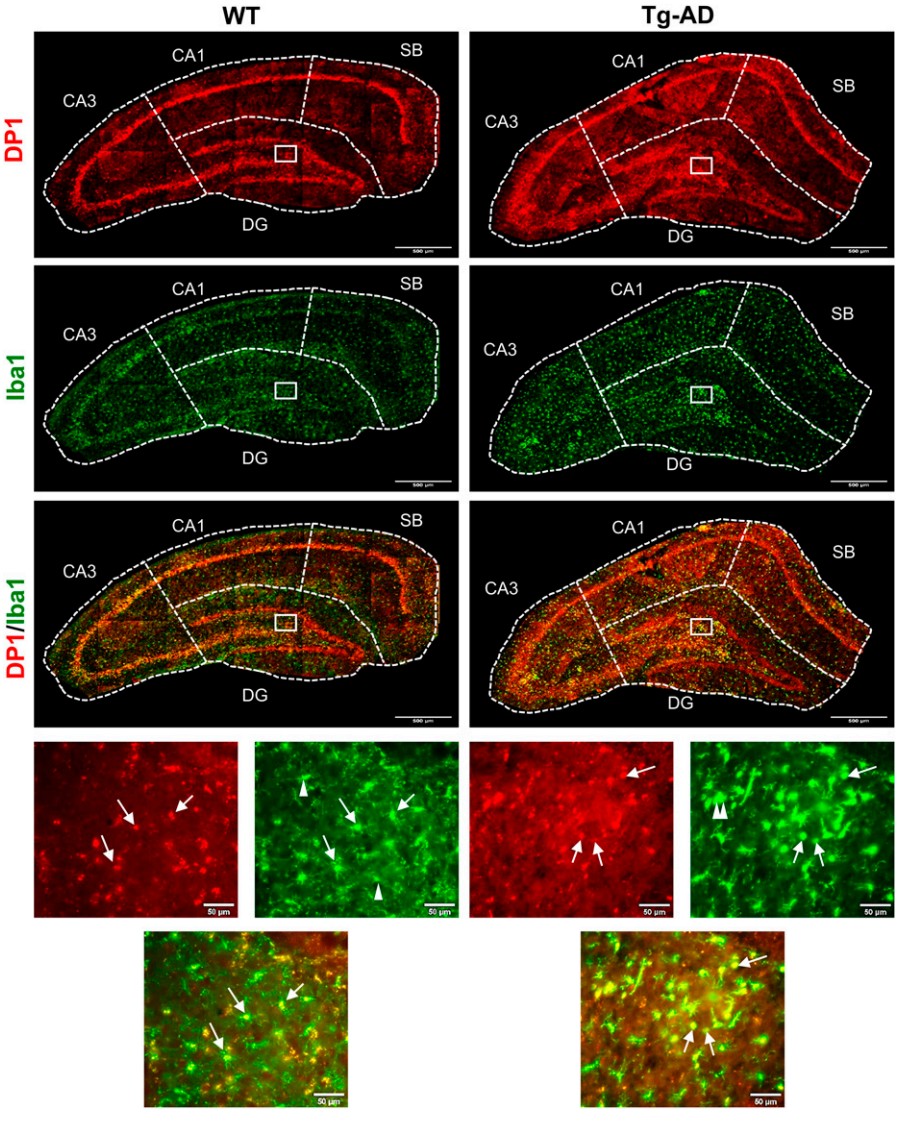

**Figure 2. Tg-AD rats have higher levels of microglia and DP1/microglia co-localization in the dorsal hippocampus than WT rats (IHC Images).**
DP1 (red), microglia (green, Iba1 antibody), and DP1/ microglia co-localization (yellow) IHC analysis of the right dorsal hippocampus of WT (left column, n = 12) and Tg-AD (right column, n = 11). Large panels: 10× magnification, 500-μm scale bars. Small (bottom) panels: 20× magnification of the small white boxes depicted in the larger panels, 50-μm scale bars. White arrows indicate the following: full, DP1/microglia co-localization; single head, ramified microglia, double head, amoeboid microglia.
Source data are available for this figure.

Finally, *amoeboid* microglia are the fewest, have the largest soma and fewest processes, and perform phagocytosis.

Notably, when compared with WT controls, Tg-AD rats showed a shift from a neuroprotective state typical of ramified microglia, to more of a neurotoxic and overactive state attributable to amoeboid microglia. In the hippocampal DG (HL), there are significant less ramified microglia (14.5% less, $t$ = 2.32, $P$ = 0.02) with a concomitant increase in reactive (1.6-fold more, $t$ = 2.52, $P$ = 0.01) and amoeboid (1.8-fold more, $t$ = 1.89, $P$ = 0.04) microglia in Tg-AD rats compared with WT controls (Fig 3).

Co-localization of DP1 with each microglia phenotype in the hippocampal DG (HL) was significantly higher in Tg-AD rats than in WT littermates (Fig 3). Accordingly, compared with WT controls, the Tg-AD rats had 1.3-fold ($t$ = 2.19, $P$ = 0.02), 2.5-fold ($t$ = 3.53, $P$ = 0.002), and 3.2-fold ($t$ = 3.08, $P$ = 0.003) higher DP1 co-localization with ramified, reactive, and amoeboid microglia, respectively.

No significant differences between WT and Tg-AD rats were detected for astrocyte levels by IHC analysis at 11 mo of age in all hippocampal regions (GFAP levels, Table S4).

## Tg-AD rats display a loss of neurons as well as lower DP2 levels within the granular cell subregion of the hippocampus

Neuronal density across hippocampal regions (SB, CA1, and CA3 pyramidal cell layers, and DG granular cell layer) were assessed with NeuN (green) to quantify mature neurons (Figs 4, 5, and S1). Similar to what was reported in the original study for Tg-AD rats at 16 and 26 mo of age (Cohen et al, 2013), we observed a significant neuronal loss (NeuN signal) though earlier, at 11 mo of age. We detected neuronal loss in Tg-AD compared with WT rats, only in the granular cell layer (GCL) and CA3c pyramidal cell layer of DG (44.2% less, t = 4.75, $P$ < 0.0001 for GCL, and 21.4% less, t = 2.07, $P$ = 0.03 for CA3c) (Fig 5, left graphs). It is clear that the thickness of the GCL is greater in WT than in Tg-AD rats, shown at higher magnification (Fig 4, bottom panels indicated by white double head arrows). Neuronal levels analyzed across all other hippocampal regions revealed no changes in Tg-AD compared with WT rats (Table S5).

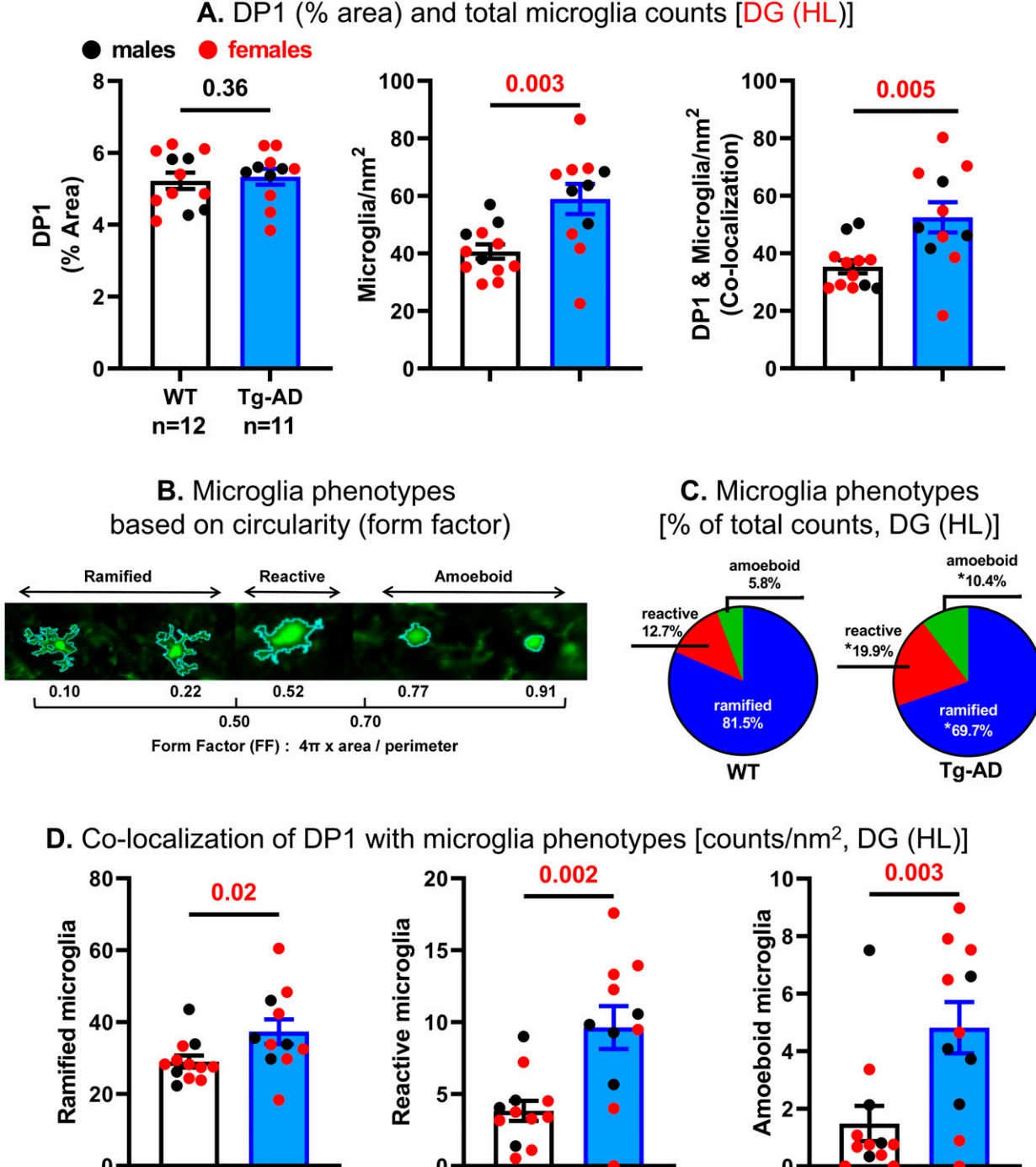

**Figure 3. Tg-AD rats have higher levels of microglia and DP1/microglia co-localization in the dorsal hippocampus than WT rats (Quantification).**
**(A)** For DP1 levels, there were no significant differences between WT and Tg-AD rats across all hippocampal regions (A, left graph for DG and Table S1). **(A)** Tg-AD rats had significantly more microglia than WT rats as shown in the hilar (HL) region (1.4-fold, $t$ = 3.15, $P$ = 0.003) (A, middle graph for DG and Table S2). **(A)** Tg-AD rats also had significantly higher levels (1.5-fold, $t$ = 2.99, $P$ = 0.005) of DP1/microglia co-localization than their WT littermates, only in the DG (HL) region (A, right graph for DG only and Table S3). **(B)** The three microglia (Iba1+) phenotypes based on circularity (form factor) as explained under material and methods. **(C)** Microglia phenotypes as % of total counts at DG (HL). Each pie slice represents the proportion of each phenotype relative to the total sum, in WT (left) and Tg-AD (right) rats. Tg-AD rats had significantly fewer ramified (14.5% less, $t$ = 2.32, $P$ = 0.02), but more reactive (1.6-fold more, $t$ = 2.52, $P$ = 0.01) and almost double amoeboid microglia (1.8-fold more, $t$ = 1.89, $P$ = 0.04) than controls. **(D)** Co-localization of DP1 with each microglia phenotype in the hippocampal DG (HL) was significantly higher in Tg-AD rats than in WT littermates (ramified: 1.3-fold $t$ = 2.19, $P$ = 0.02; reactive: 2.5-fold [$t$ = 3.53, $P$ = 0.002], and; ameboid: 3.2-fold $t$ = 3.08, $P$ = 0.003). Significance ($P$-values shown on graphs) estimated by a one-tailed Welch's $t$ test.

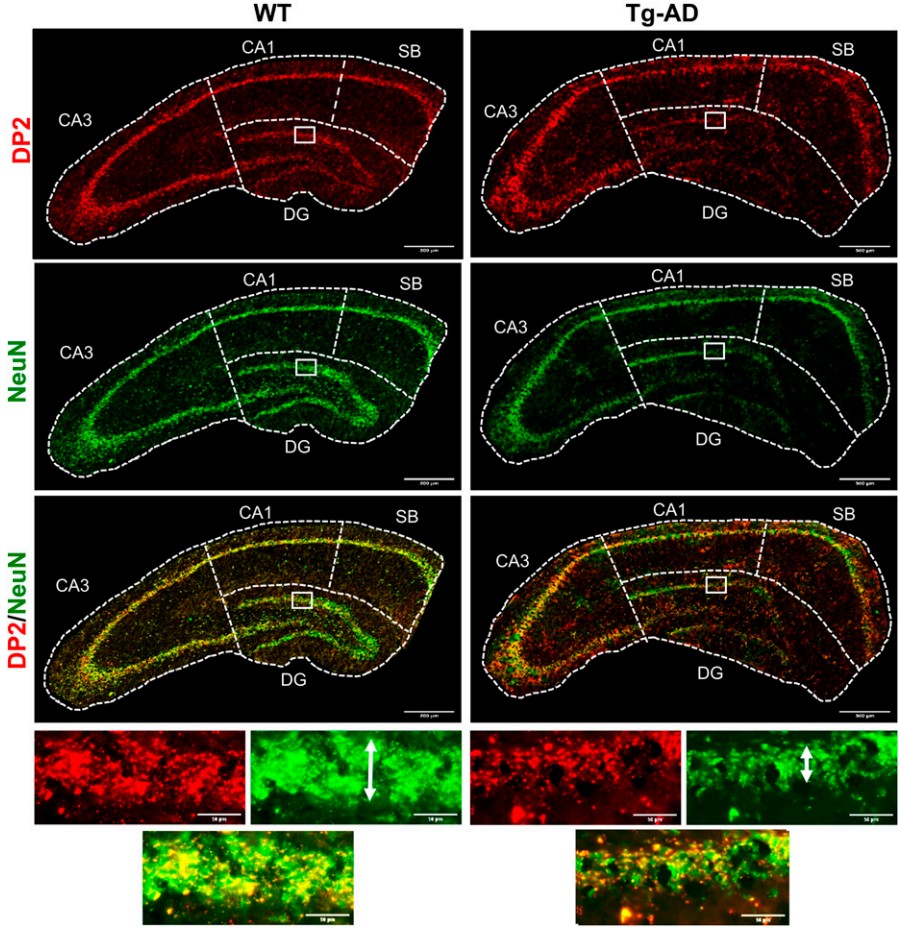

**Figure 4. Tg-AD rats display DP2 and neuronal losses in the dorsal hippocampus (IHC Images).**
**(A)** DP2 (red), neurons (green, NeuN antibody), and DP2/neuronal co-localization (yellow) IHC analysis of the right dorsal hippocampus of WT (left column, n = 12) and Tg-AD (right column, n = 11). Large panels: 10× magnification, 500-μm scale bars. Small (bottom) panels: 20× magnification of the small white boxes at the GCL depicted in the larger panels, 50-μm scale bars. It is clear that the thickness of the GCL is greater in WT than in Tg-AD rats (white double head arrows). Source data are available for this figure.

A similar trend was detected for DP2 receptor levels (red) at the GCL only (Figs 4 and 5). Tg-AD rats exhibited significantly fewer DP2 receptors in GCL than the WT controls (34.4% less, $t$ = 7.25, $P$ < 0.0001 for GCL) (Fig 5 and Table S6). In all other hippocampal regions, there were no differences in DP2 levels between WT and TG-AD rats, except in the CA1 region, where DP2 levels were 1.3-fold higher in Tg-AD than in WT controls ($t$ = 3.36, $P$ = 0.002, Fig 5 and Table S6). The observed DP2 increase in the CA1 region of Tg-AD rats is likely due to the presence of Aβ plaques.

It is clear that at least 50% of DP2 is co-localized with neurons, as shown in Fig 4 (yellow). For this reason, DP2 receptor and neuronal co-localization was not significantly different between WT and Tg-AD rats at all hippocampal regions. This finding supports that at least 50% of NeuN and DP2 signals are co-localized and that their decrease in Tg-AD compared with WT, follows the same trend (Table S7).

### Lipocalin prostaglandin D2 synthase (L-PGDS) mRNA levels are the highest among 33 genes evaluated by RNAseq in the hippocampus of WT and Tg-AD rats

We assessed the mRNA levels for 33 genes involved in the PGD2 and PGE2 pathways in hippocampal tissue from WT and Tg-AD male (five of each genotype) and female (five of each genotype) rats. The RNAseq analysis reports output measures as reads per million (RPM), as well as false discovery rate (FDR) and $P$-values (Table S8).

In addition, the mRNA levels (mean RPMs) for 28 of those genes in WT females only are displayed in 2 bar graphs, one representing the top 14 most abundant genes (Fig 6) and the other including the 14 least abundant genes (Fig 6). The gene groups within the PG pathway are depicted in each of the bar graph. There were no significant genotype (WT versus Tg-AD) or sex (male versus female) differences in the expression levels for most of these genes (Figs S2–S6 including each gene nomenclature, and Table S8).

Overall, the data revealed that mRNA transcript levels were the highest for L-PGDS, the PGD2 synthase in the brain (RPM = 282.7, Fig 6). This is consistent with PGD2 being the most abundant prostanoid of the four that we measured in hippocampal tissue (as much as 70% of the total, Fig 1). Hematopoietic-PGDS (H-PGDS), the other PGD2 synthase that is mainly detected in microglia (Mohri et al, 2007), was minimally expressed (RPM = 1.13, Fig 6). There were three PGE2 synthases detected by RNAseq, and their mean RPMs were in descending order, 165.8 (PGES-3), 46.5 (PGES-2), and 16.5 (PGES-3-like1) (Fig 6).

Evaluation of four genes involved in PG biosynthesis and metabolism (Fig 6), showed that prostaglandin reductase-2 (pTGR-2), which metabolizes PGs, exhibited the highest expression (RPM = 70.6). The remaining three genes followed in descending order, COX-2 (RPM = 23.6), COX-1 (RPM = 16.6), and phospholipase A2 (RPM = 7.4).

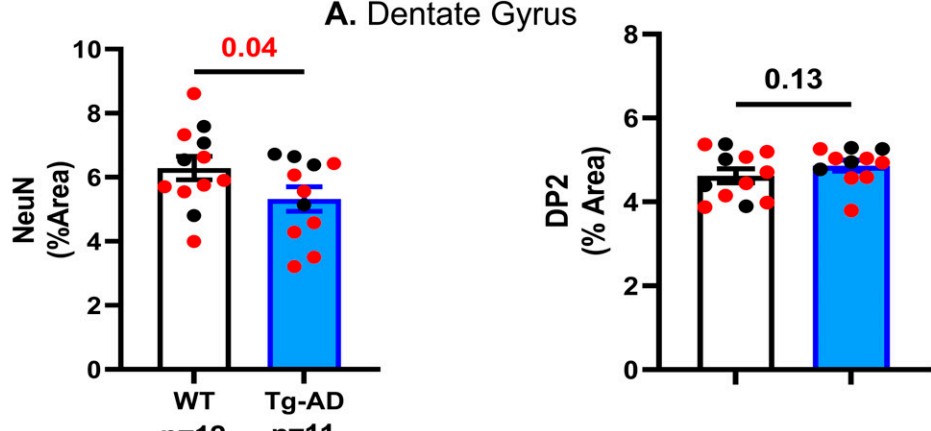

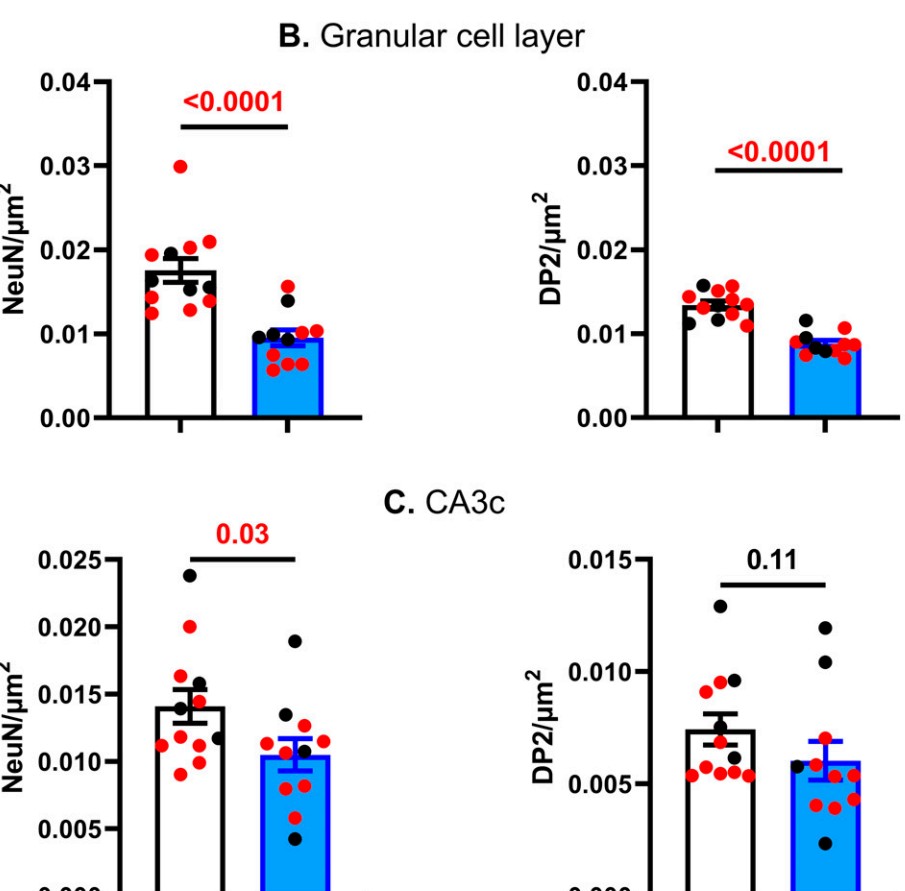

**Figure 5. Tg-AD rats display DP2 and neuronal losses in the dorsal hippocampus (Quantification).**
**(A, B, C)** Neuronal loss (left graphs) detected only in the DG (A), at the GCL (B) and CA3c (C) of Tg-AD compared with WT rats (44.2% less, t = 4.75, *P* < 0.0001 for GCL, and 21.4% less, *t* = 2.07, *P* = 0.03 for CA3c). Neuronal density analyzed across all other hippocampal regions revealed no changes in Tg-AD compared with WT rats (Table S5). **(A, B, C, right graphs)** DP2 loss detected only at the GCL (34.4% less, *t* = 7.25, *P* < 0.0001 for GCL) (B) but not at the other hippocampal locations (A and C, and Table S6) of Tg-AD compared with WT rats. Significance (*P*-values shown on graphs) estimated by a one-tailed Welch's *t* test.

In the rat hippocampal tissue, the mRNA levels (RPM) for PGD2 receptors were, in descending order, as follows (Fig 6): DP1 (rat, orthologous to human DP1, 0.43), DP2 (0.41), and DP1 (0.09). The rat genome has two DP1 copies (genes: *PTGDR*, ID: 63889 and *PTGDRL*, ID: 498475). The protein alignments are highly similar (354/357 residues, 99% homology, NCBI groups the two in an identical protein group), differing only on their location on chromosome 15. RPMs for PPARγ, a putative PGJ2 receptor, were = 0.24. Notably, PPARγ activators were expressed at higher levels than the receptor

itself (Fig 6): co-activator related 1 (30.3), co-activator 1α (20.2), and co-activator 1β (6.6). The receptors for PGE2 showed the highest RPM levels (Fig 6) listed in descending order: EP1 (5.51), EP3 (1.54), EP2 (1.14), and EP4 (0.39).

The SRY-box transcription factor 2 (Sox-2) gene is a transcription factor best known as a reprogramming factor necessary for generating induced pluripotent stem cells (Takahashi & Yamanaka, 2006). Sox-2 is also required for proliferation and differentiation of oligodendrocytes during postnatal brain myelination and CNS

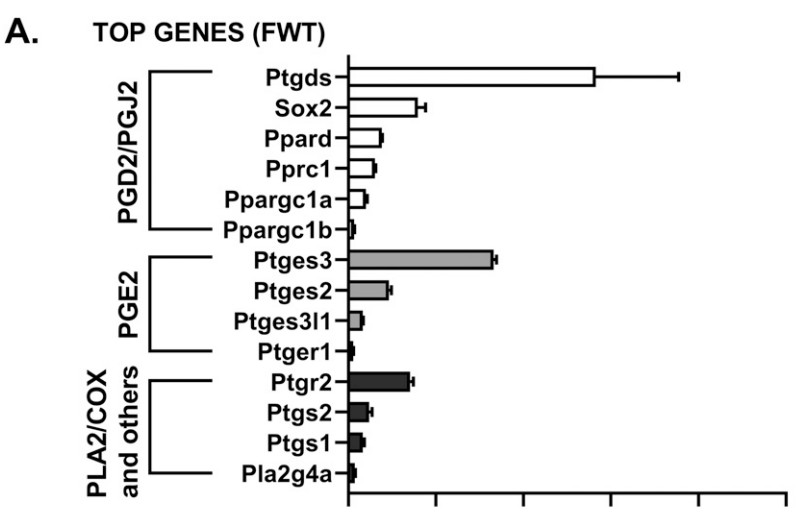

## A. TOP GENES (FWT)

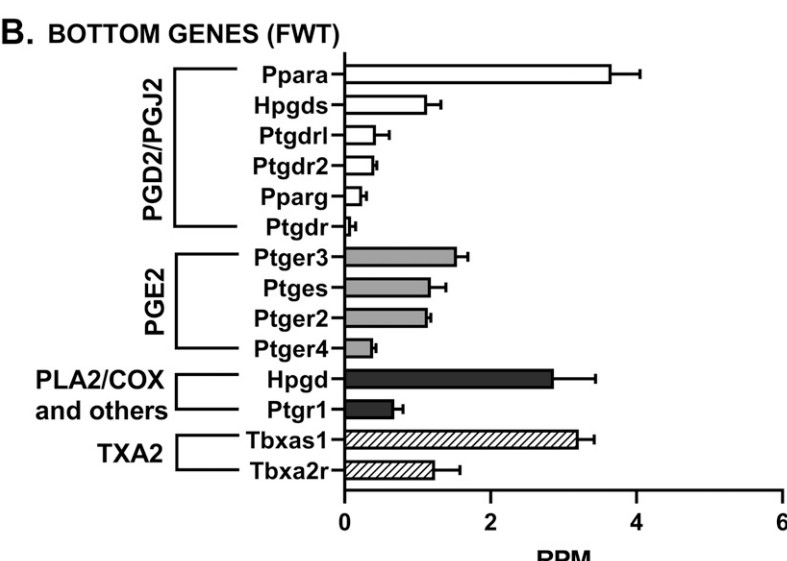

## B. BOTTOM GENES (FWT)

**Figure 6. Lipocalin prostaglandin D2 synthase (L-PGDS) mRNA levels are the highest among 28 genes evaluated by RNAseq in the hippocampus of WT and Tg-AD rats.**

mRNA levels for 28 genes involved in the PG pathway were determined by RNAseq in whole left hippocampal tissue (combined ventral and dorsal) from 11-mo WT and Tg-AD male (five of each genotype) and female (five of each genotype) rats. **(A, B)** mRNA levels (mean RPMs ± SEM) for 28 of those genes in WT females only are displayed in 2 bar graphs, one representing the 14 most abundant genes (A), and the other including the 14 least abundant genes (B). The gene groups within the PG pathway are depicted in each bar graphs. **(A)** The mRNA transcript level was the highest for L-PGDS, the PGD2 synthase in the brain (A). Most of the mRNA levels of the 28 genes were not significantly different between Tg-AD rats and WT littermates, except, for example, the transcription factor Sox-2, which was significantly down-regulated in male Tg-AD rats compared with their WT littermates. Additional details are in the text and Table S8. mRNA levels (mean RPMs ± SEM) for the other experimental groups are displayed in Figs S2–S6, including each gene nomenclature.
Source data are available for this figure.

remyelination (Zhang et al, 2018). In addition, Sox-2 is a negative regulator of myelination by Schwann cells (Florio et al, 2018), and its levels are controlled by L-PGDS in PNS injured nerves (Forese et al, 2020). We found that Sox-2 expression levels in the hippocampal tissue were quite high (79.4 RPM, Fig 6), being the third highest expressed gene in our list, after L-PGDS and PGES-3 (Table S8). Moreover, Sox-2 was significantly down-regulated in male Tg-AD rats compared with their WT littermates (22.2% less, *P* = 0.011, Table S8). The significance of these data will be addressed in the discussion.

Western blot analyses for six proteins involved in the PGD2 pathway, that is, receptors DP1, DP2, and PPARγ, the synthase L-PGDS, as well as COX-2 and Sox-2, are shown in Fig S7. The data indicate no changes in the levels of these protein in hippocampal tissue from WT and Tg-AD male (n = 3 for each genotype) and female (n = 3 for each genotype) rats (Table S9). The whole image of the Western blots for DP1, DP2, and L-PGDS is shown in Fig S8.

## Tg-AD rats show enhanced FL-APP and Aβ peptide levels as well as Aβ plaques in the hippocampus

The original study by Cohen et al reported that Tg-AD rats express 2.6-fold higher levels of human full-length APP than their WT littermates, assessed by Western blot analysis of the brain (Cohen et al, 2013). Full-length APP (FL-APP) was detected with the mouse monoclonal antibody 22C11, which reacts with human and rat, as well as other species (manufacturer's specifications). In our studies using the same antibody, it is clear that the levels of FL-APP are 5.6-fold higher in the hippocampal tissue of Tg-AD than WT rats (Fig 7, top panels labeled with FL-APP, and Fig 7, left graph, combined males and females, *t* = 7.23, *P* < 0.001). This trend was observed in males (n = 3 for each genotype) and females (n = 3 for each genotype), and the values were normalized for actin (Fig 7, second panels). The higher levels of FL-APP detected in our analysis compared with those detected in the Cohen et al (2013) analysis,

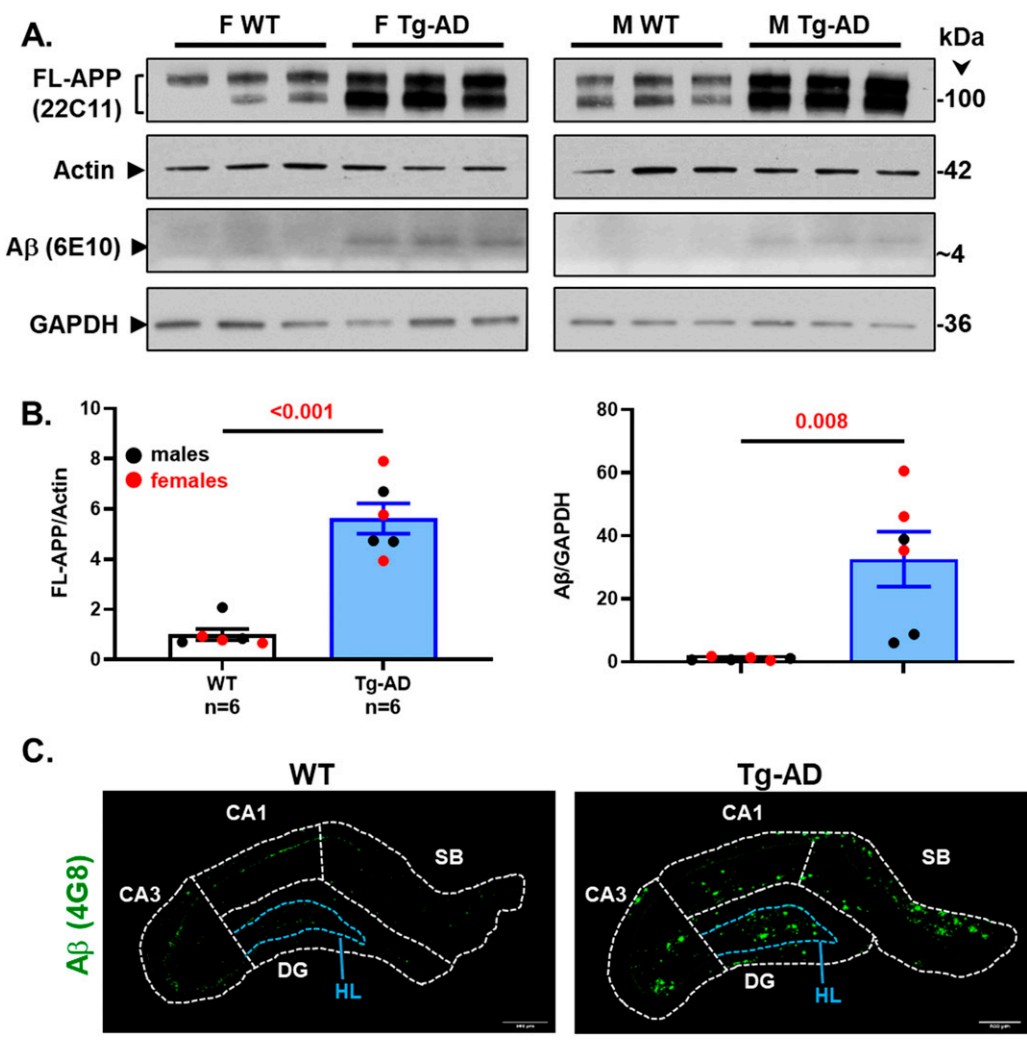

**Figure 7. Tg-AD rats show enhanced FL-APP and Aβ peptide levels as well as Aβ plaques in the hippocampus.**
**(A)** FL-APP (top panels) and Aβ levels (third panels from the top) were assessed by Western blot analysis in whole left hippocampal (combined ventral and dorsal) homogenates from 11-mo WT and Tg-AD male (M, three of each genotype) and female (F, three of each genotype) rats. Actin (second panels from the top) and GAPDH (bottom panels) detection served as the respective loading controls. **(B)** FL-APP and Aβ levels are significantly higher in Tg-AD rats compared with WT littermates as semi-quantified by densitometry (FL-APP: $t = 7.23$, $P < 0.001$; Aβ: $t = 3.62$, $P = 0.008$). Data represent the percentage of the pixel ratio for FL-APP and Aβ over the respective loading controls for Tg-AD compared with WT (represented as a value of one). Values are means ± SEM from six rats per genotype (males and females combined). Significance (*P*-values shown on graphs) estimated by a one-tailed Welch's *t* test. **(C)** Immunohistochemistry for Aβ plaque load for WT (left panel) and Tg-AD (right panel) rats is shown at 10× magnification, scale bars = 500 *μm*. All Tg-AD rats used in this study exhibited Aβ plaques, but not their WT littermates.
Source data are available for this figure.

could be explained by our studies using hippocampal tissue while whole brain tissue was used in the Cohen studies.

We also assessed Aβ levels in the same samples of rat hippocampal tissue with the mouse monoclonal antibody 6E10, which has a threefold higher affinity for human APP and Aβ compared with rat (manufacturer's specifications). Aβ peptides were detected in male and female Tg-AD rats but not in the WT littermates, as shown in Fig 7 (third panels labeled with Aβ [6E10]), and semi-quantified in Fig 7 (right graph, combined males and females, $t = 3.62$, $P = 0.008$). The whole images of the Western blots for FL-APP and Aβ are shown in Fig S9.

The presence of Aβ plaques in all 11-mo Tg-AD rats was confirmed by IHC analysis with the mouse monoclonal antibody 4G8, as shown for a female rat in Fig 7, right panel. A WT female rat is included for comparison (Fig 7, left panel).

**Tg-AD rats exhibit impaired spatial learning and memory**

In the original studies with the Tg-AD rats (Cohen et al, 2013), cognitive behavior was assessed at 6, 15, and 24 mo of age. Most of the significant changes between WT and Tg-AD rats were detected at 15 and 24 mo, but not at 6 mo of age (Cohen et al, 2013). To shorten the experimental time line, we evaluated the WT and Tg-AD rats at an earlier age, that is, at 11 mo. We evaluated cognitive impairment with two hippocampal-dependent tasks to measure short-term learning/memory and navigation: Radial 8-arm maze (RAM), which is a passive behavioral task, and the active-place avoidance task (aPAT). Because in the original studies no sex differences were reported (Cohen et al, 2013), males and females were combined for our analyses.

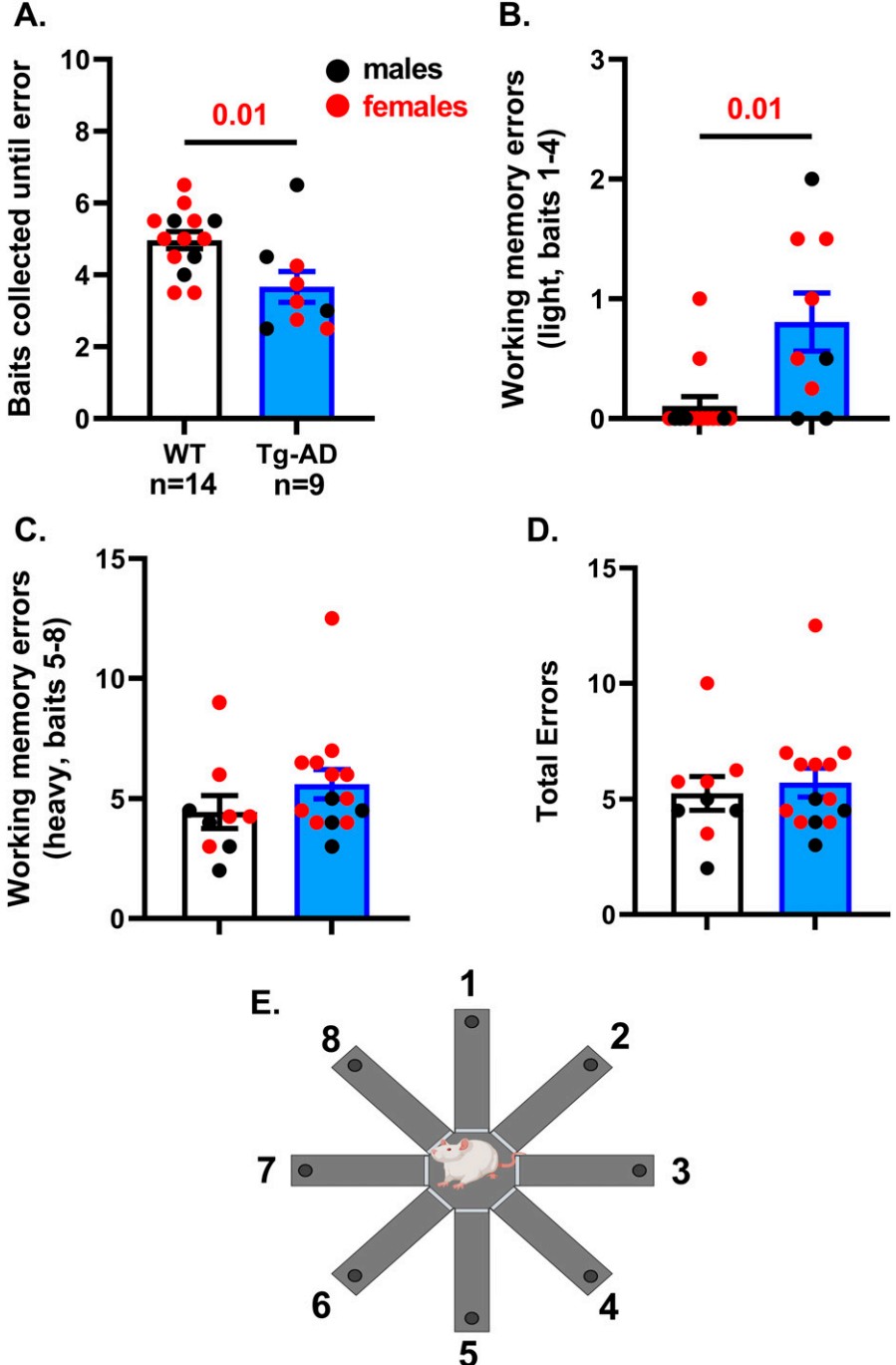

**Figure 8. Tg-AD rats exhibit impaired spatial learning and memory at 11 mo of age in the 8-Arm Radial Arm Maze.**
**(A, B)** Results with RAM show Tg-AD rats (n = 9) commit significantly more errors versus WT rats (n = 14) when analyzed (A) for baits collected until first error ($t$ = 2.65, $P$ = 0.01), and (B) for working memory errors committed during collection of bait numbers 1–4 (light memory load) ($t$ = 2.75, $P$ = 0.01). **(C, D)** No significant differences were observed between conditions with a heavy (challenging) working memory load (collection of baits 5–8) ($t$ = 1.26, $P$ = 0.11) or (D) for total errors committed collecting all eight baits ($t$ = 0.48, $P$ = 0.32). Significance estimated with a one-tailed Welch's $t$ test, and $P$-values are shown above bar graphs. **(E)** RAM with the arms labeled 1–8 (created with BioRender.com).
Source data are available for this figure.

In our RAM studies, we wanted to assess total errors made, baits collected until error, and working memory. Two forms of working memory were assessed, a light working memory load for baits 1–4, and a heavy (challenging) working memory load for baits 5–8 (Fig 8). For RAM, the rat groups were WT (10 females, 4 males) and Tg-AD (5 females and 4 males). We found that Tg-AD rats had a behavioral deficit in outputs for baits collected until error (Fig 8, $t$ = 2.65, $P$ = 0.01) and in light working memory load (Fig 8, $t$ = 2.75, $P$ = 0.01). We found no differences in the more difficult measures such as heavy

working memory load (Fig 8; $t$ = 1.26, $P$ = 0.11) and total errors (Fig 8, $t$ = 0.48, $P$ = 0.32). These findings show a working memory impairment in the early and less challenging part of the task for the Tg-AD rats compared with controls.

In the aPAT analysis, we used a separate cohort of rats. In aPAT, the rat groups were WT (seven females, seven males) and Tg-AD (eight females, six males). We found a significant deficit in spatial reference memory during training for the Tg-AD rats in all of the reported measures: latency to first entrance (Fig 9, $F_{(1,26)}$ = 5.73,

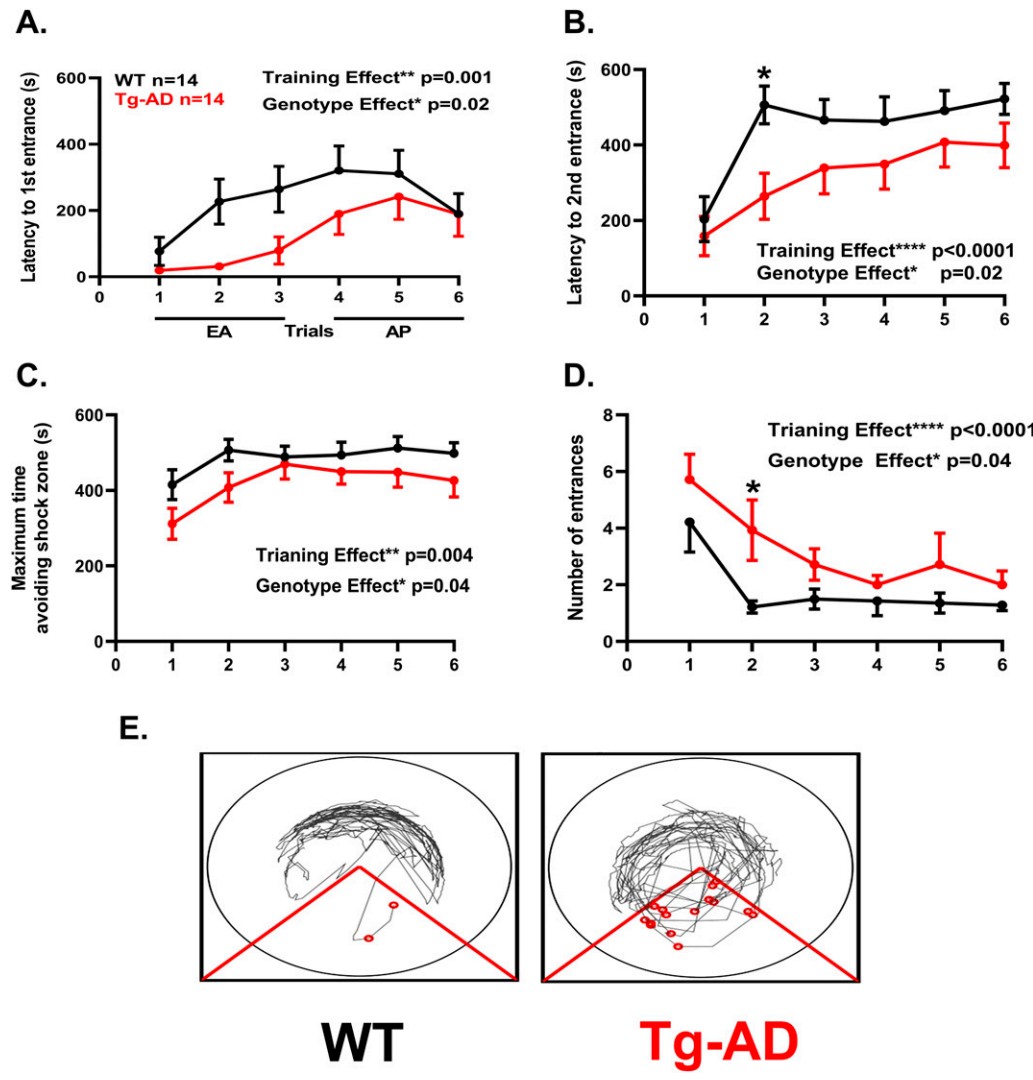

**Figure 9. Tg-AD rats exhibit impaired spatial learning and memory performance at 11 mo of age in the active place avoidance task.**
All behavioral measures with aPAT show overall effects by genotype and by trials indicated as genotype and treatment, respectively, with corresponding *P*-values. **(A)** Analysis of latency to first entrance shows Tg-AD rats exhibiting significantly shorter latencies ($F_{(1,26)}$ = 5.73, *P* = 0.02). **(B)** Latency to second entrance shows a significant main effect and a post hoc difference on trial 2, ($F_{(1,26)}$ = 5.70, *P* = 0.02, trial 2: *P* = 0.03). **(C)** Maximum time to avoid shows Tg-AD rats exhibiting significantly shorter maximum avoidance latencies ($F_{(1,26)}$ = 4.74, *P* = 0.04). **(D)** Number of entrances shows a significant main effect and a post hoc difference on trial 2 (*t* = 2.81, *P* = 0.03, trial 2: *P* = 0.03). **(E)** Representative track tracings for trial 2 shown for WT and Tg-AD female rats. Significance estimated by a two-way repeated measure ANOVA with Sidak's post hoc for multiple comparisons.
Source data are available for this figure.

*P* = 0.02), latency to second entrance (Fig 9, $F_{(1,26)}$ = 5.70, *P* = 0.02), maximum time avoidance (Fig 9, $F_{(1,26)}$ = 4.74, *P* = 0.04) and entrances (Fig 9, $F_{(1,26)}$ = 4.78, *P* = 0.04). Significant post hoc differences were observed at trial 2 during the early acquisition (EA) phase in latency to second entrance (Fig 9, *t* = 3.05, *P* = 0.03), and in the number of entrances (Fig 9, *t* = 2.81, *P* = 0.03). Representative track tracings for trial 2 are shown for WT and Tg-AD female rats (Fig 9).

## Timapiprant improves spatial learning and mitigates plaque burden, neuronal loss, and microgliosis in Tg-AD rats

We used the hippocampal-dependent active place avoidance task to assess short-term working memory performance on 11-mo-old Tg-AD non-treated (TGNT) and Tg-AD timapiprant-treated (TGTR) male rats, and the equivalent WT male littermates. The measurement latency to first entrance into the shock zone for TGTR versus TGNT revealed an overall significant effect of treatment (Fig 10, $F_{(1, 16)}$ = 13.87, *P* = 0.002) and of training (Fig 10, $F_{(5, 80)}$ = 2.93, *P* = 0.02). Significant post hoc differences were observed at trial 5 during the asymptotic performance (AP) phase in latency to first entrance (Fig 10, *t* = 3.15, *P* = 0.01) between TGTR and TGNT rats. No differences were detected between WTNT and WTTR males ($F_{(1, 16)}$ = 1.04, *P* = 0.32), nor between WTNT and TGTR ($F_{(1, 16)}$ = 0.88, *P* = 0.36), showing the beneficial effects of timapiprant-treatment only under pathological conditions. Graphed data (10C-10E) represent a ratio of the Tg-AD rats over their WT controls.

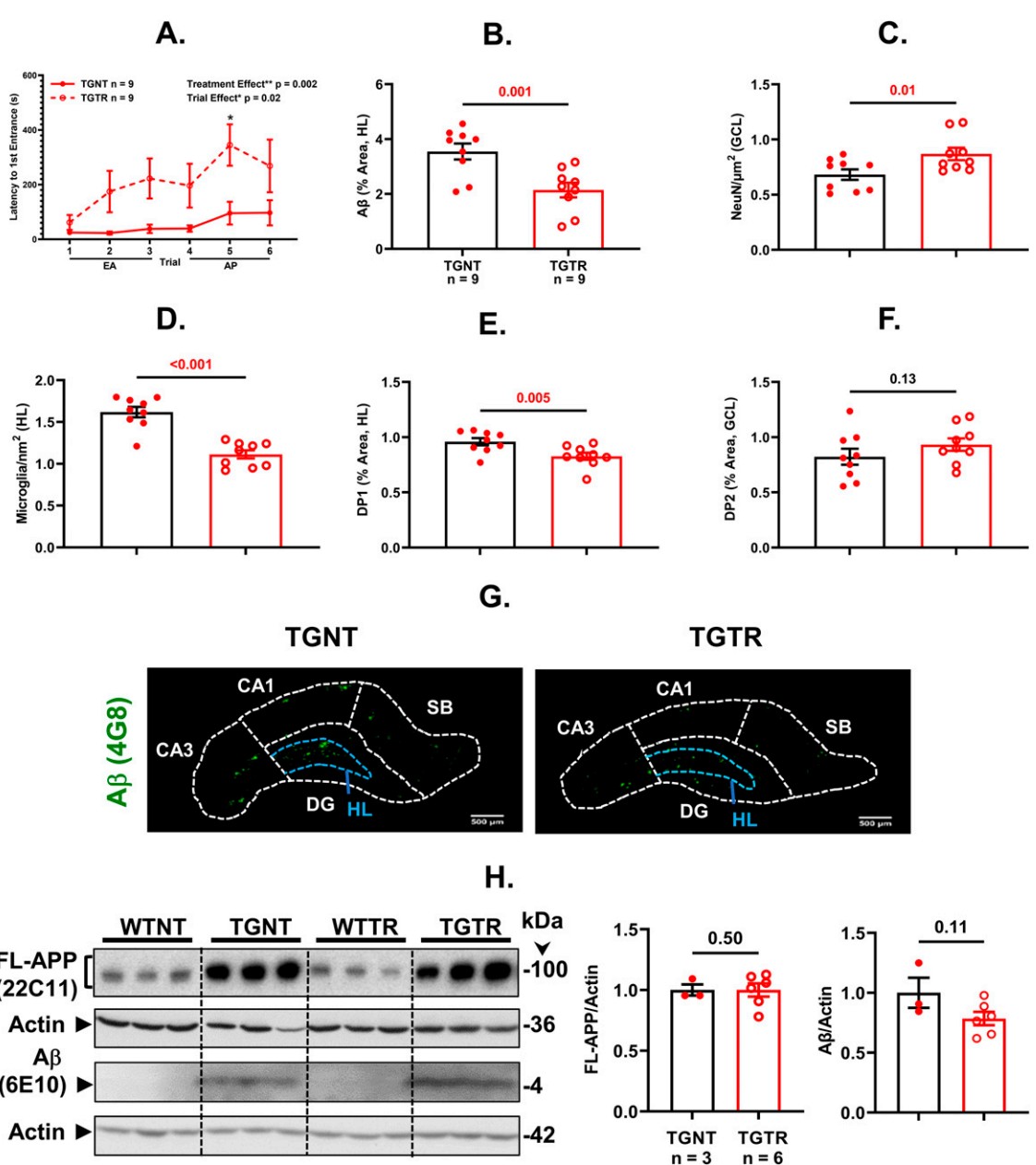

**Figure 10. Timapiprant mitigates AD pathology.**
**(A, B, C, D)** Tg-AD timapiprant-treated (TGTR, n = 9) compared with Tg-AD non-treated (TGNT, n = 9) male rats perform significantly better in latency to first entrance during training (A, with a significant post hoc difference on trial 5 in the asymptotic performance stage) ($F_{(1, 16)}$ = 13.87, P = 0.002, trial 5: P = 0.02), experience lower plaque burden (B, t = 3.55, P = 0.001), higher neuronal levels (C, t = 2.56, P = 0.01), and lower microglia levels (D, t = 6.37, P < 0.001) in the DG hilar subregion of the hippocampus. **(E, F)** DP1 and DP2 receptor levels were decreased (E, t = 2.78, P = 0.007) and unchanged (F, t = 1.29, P = 0.11), respectively, in the same hippocampal subregion of TGTR versus TGNT male rats. Unpaired one-tail t tests with Welch's corrections were used for statistical analysis. EA, early acquisition; AP, asymptotic performance; (*P < 0.05; **P < 0.01; ***P < 0.001). **(G)** Immunohistochemistry for Aβ plaque load for TGNT (left panel) and TGTR (right panel) rats is shown at 10× magnification, scale bars = 500 $\mu$m. **(H)** FL-APP (top panels) and Aβ levels (third panels from the top) were assessed by Western blot analysis in whole left hippocampal (combined ventral and dorsal) homogenates from 11-mo WT and Tg-AD (TG) not treated (NT) and timapiprant treat (TR) male rats and no significant differences were seen (t = 2.34, P = 0.50). Actin (second and fourth panels from the top) detection served as loading control. Graphs show FL-APP (left) and Aβ (right) levels semi-quantified by densitometry. Data represent the percentage of the pixel ratio for FL-APP and Aβ over the respective loading controls for TGTR compared with TGNT (represented as a value of one). **(C, D, E, F, H)** All graphed data (C, D, E, F, and H except for Aβ/actin) represent a ratio of the Tg-AD over their WT controls. Values are means ± SEM. Significance (P-values shown on graphs) estimated by a one-tailed Welch's t test.
Source data are available for this figure.

We compared Aβ plaque burden in the hippocampal DG hilar subregion between TGTR and TGNT rats and found that timapiprant significantly mitigated Aβ plaque load (Fig 10, $t$ = 3.55, $P$ = 0.001, and Fig 10). Similarly, timapiprant alleviated neuronal loss in the GCL subregion (Fig 10, $t$ = 2.56, $P$ = 0.01) and microgliosis in the hilar subregion (Fig 10, $t$ = 6.37, $P$ < 0.001) for TGTR compared with TGNT male rats. DP1 receptor levels were decreased in the DG hilar subregion of TGTR compared with TGNT male rats (Fig 10, $t$ = 2.78, $P$ = 0.007), but not those of the DP2 receptor in the GCL (Fig 10, $t$ = 1.29, $P$ = 0.11). The hippocampal IHC images for DP1/Iba1 and DP2/NeuN as well as the full westerns for DP1, DP2, and L-PGDS from TGNT and TGTR male rats are shown in Fig S10.

The levels of FL-APP and Aβ were similar in TGNT and TGTR males when assessed by Western blot analysis (Fig 10, $t$ = 2.34, $P$ = 0.50). The discrepancy on the Aβ level comparison between TGNT and TGTR rats is explained by the Western blot analysis (Fig 10) with the whole hippocampus, whereas the IHC analysis (Fig 10) includes Aβ plaques only in the hippocampal DG hilar subregion. The whole images of the Western blots for FL-APP and Aβ are shown in Fig S11.

## Discussion

There is still much to learn about the profile and role of PGs in AD pathology. We focused on the PGD2 pathway because PGD2 is the most abundant PG in the brain, and its contribution to AD merits more attention. Much more is known about the relation between the PGE2 pathway and AD (Woodling & Andreasson, 2016). Investigating the relevance of the PGD2 pathway to AD could discover potential biomarkers and/or therapeutic targets for treating this devastating disease.

We investigated the PGD2 pathway in TgF344-AD (Tg-AD) transgenic rats at 11 mo of age because it is midway between the ages at which these rats present mild (at 6 mo of age) and robust (at 16 mo of age) AD pathology, as reported in the original study (Cohen et al, 2013). Understanding pre- and/or early-stages of AD is paramount, as treating AD at these stages would be the best approach to preventing severe progression (Sun et al, 2018). We established that at 11 mo of age, the Tg-AD rats exhibit impaired hippocampal-dependent spatial learning and memory, as well as molecular markers of AD, such as amyloid plaques, microglial activation, neuronal loss, and early signs of τ-PHF, the latter reported by us (Oliveros et al, 2022).

Interestingly, we found that neuronal loss at 11 mo of age was specific to the DG and its subregions GCL and CA3c in the hippocampal tissue. The DG is known to be vulnerable to aging and to be affected in the early stages of AD (Takeda & Tamano, 2018). In fact it is reported that in AD the GCL of the DG has impaired firing (Palmer & Good, 2011). When the GCL is impaired, the ability to identify/discriminate environmental cues during memory formation is greatly diminished in spatial learning/memory (Lee & Jung, 2017). These data support our findings that at 11 mo of age, Tg-AD rats exhibit a significant impairment in two separate hippocampal-dependent behavioral tasks where the use of environmental cues is necessary to evaluate learning/memory.

The relative abundance of the four prostanoids that we measured in hippocampal tissue was, in descending order, PGD2, TxB2,

PGE2, and PGJ2. PGD2 was by far the most abundant at ~46.2 pg/mg wet tissue. PGD2 was ~3-fold higher than TxB2, ~16-fold higher than PGE2, and ~28-fold higher than PGJ2, all reported as an average between WT and Tg-AD rats because there was no significant difference between the two genotypes. Other studies confirm our finding that PGD2 is the most abundant PG in the brain, including in human brains (Ogorochi et al, 1984; Hertting & Seregi, 1989; Ricciotti & FitzGerald, 2011; Shaik et al, 2014). Alternatively, studies using radioimmunoassays to measure PGD2 levels in brains of male Wistar rats killed by microwave irradiation reported significantly lower PGD2 levels, such as 2.3 pg/mg wet tissue, thus almost 20-fold less than what we found (Narumiya et al, 1982). This discrepancy can be attributed to the different methodology used, radioimmuno-assays in the older studies (Narumiya et al, 1982) versus quantitative LC–MS/MS in the most recent studies (Shaik et al, 2014) and our studies. LC–MS/MS exhibits superior sensitivity, accuracy, efficiency, and lack of cross-reactivity compared with radioimmuno-assays (Brose et al, 2011, 2013; Dong et al, 2018).

PGD2 levels were equivalent in Tg-AD and WT rats. In contrast, one study reported that PGD2 levels were significantly higher in postmortem frontal cerebral cortex tissue from AD patients compared with age-matched controls (Iwamoto et al, 1989). Moreover, others demonstrated that PGD2 levels increase significantly by as much as sixfold in the hippocampal and/or cerebral cortical tissue of male Sprague–Dawley rats after traumatic brain injury (Kunz et al, 2002) or brain ischemia (Liu et al, 2013a, 2013b; Shaik et al, 2014). Several factors could explain why the levels of PGD2 were equivalent in the hippocampus of 11-mo Tg-AD and WT rats. First, the studies with the human AD cases (Iwamoto et al, 1989) measured PGD2 levels in a different brain area, that is, the cerebral cortex. Second, PG levels were assessed in the human cerebral cortices upon a 30-min incubation of the tissue at 37°C, thus assessing PGD2 produced de novo during those 30-min. Under these conditions, PGD2 levels were significantly (~2-fold) higher in the AD cases than in controls. In contrast, there were no marked differences for PGE2 between the AD cases and controls (Iwamoto et al, 1989). Third, the short half-life of PGD2 could explain the discrepancy between our studies and those involving different forms of rat brain injury. The half-life of PGD2 in mice was estimated to be 1.6 min in the brain and 1.5 min in the blood (Suzuki et al, 1986). Therefore, the increase in the levels of PGD2 under a chronic condition, such as in AD and measured in our studies, could be harder to detect than soon after brain injury such as that induced by traumatic brain injury (Kunz et al, 2002) or brain ischemia (Liu et al, 2013a, 2013b; Shaik et al, 2014). Although the exact cause of the PGD2 increase is unclear, PGD2 production could be accelerated to compensate for neuronal damage, and possibly enhance neuronal activity in the injured brain, as suggested by Iwamoto et al (1989). Clearly, further investigation into this matter is needed.

The biologic actions of PGD2 are elicited through binding to its receptors DP1 and DP2 on specific cell types. In the brain, DP1 was detected in microglia (Mohri et al, 2006), astrocytes (Mohri et al, 2006), and neurons (Liang et al, 2005). Moreover, DP1 was specifically localized in microglia and reactive astrocytes associated with senile plaques in the cerebral cortex of AD patients and of Tg2576 mice, a model of AD (Mohri et al, 2007). In our studies with 11-mo Tg-AD and WT rats, we detected changes in hippocampal microglial

numbers but not in astrocytes, thus we investigated DP1 distribution among the three microglia phenotypes, that is, ramified, reactive and amoeboid. Notably, in the hippocampal hilar subregion, Tg-AD rats had significantly fewer ramified but more reactive and almost double amoeboid microglia than controls. It is well agreed that microglia form factor is directly related to its function. The shift away from a highly branched ramified state is indicative of microglia changing in response to pathological conditions (Karperien et al, 2013). Thus, the Tg-AD rats at 11 mo of age exhibited a shift from a neuroprotective state typical of ramified microglia, to more of a neurotoxic and overactive state attributable to amoeboid microglia. This was expected as the latter state is associated with neurodegeneration (Block et al, 2007).

We established that DP1/microglia co-localization at the hippocampal hilar subregion increased the most (3.2-fold) in amoeboid microglia of Tg-AD rats compared with controls. We propose that enhanced DP1/amoeboid microglia co-localization is an early marker of neurodegeneration. In fact, microglial overactivation and recruitment are induced by A$\beta$, leading to microglia clustering around A$\beta$ aggregates at an early stage before neuropil damage in AD patients (Block et al, 2007). Furthermore, microglia-mediated neurotoxicity manifested by the production of pro-inflammatory cytokines such as TNF$\alpha$ and IL-1$\beta$, reactive oxygen/nitrogen species, and chemokines (Ahmad et al, 2017), tends to be progressive potentially contributing to the progressive nature of AD (Block et al, 2007).

Whether the increase in DP1/amoeboid microglia co-localization contributes to the neurodegenerative process or is a compensatory mechanism remains to be established. Both DP1 agonists and antagonists can be protective in the brain and/or spinal cord, depending on the type of injury (Fig 11). On the one hand, DP1 agonists such as BW245C protect against glutamate toxicity and ischemic stroke induced in rodents (Liang et al, 2005; Ahmad et al, 2010). The benefits of DP1 activation are mediated by increased cAMP synthesis that is instrumental in converting pro-inflammatory neurotoxic microglia towards a tissue reparative anti-inflammatory phenotype (Ghosh et al, 2016). Among other effects, DP1 activation facilitates vasodilation, thus protecting the brain from ischemic stroke caused by brain blood vessels becoming clogged (Ahmad et al, 2010). DP1 activation also regulates sleep by stimulating adenosine formation and subsequently activating the adenosine receptor A2A (Ahmad et al, 2019). Studies with mice showed that sleep drives A$\beta$ clearance from the adult brain (Xie et al, 2013). Both ischemic stroke (Vijayan et al, 2017) and sleep dysregulation (Kang et al, 2017) facilitate the progression of AD pathology. On the other hand, by limiting bleeding in mice, DP1 antagonists such as laropiprant (MK-0524) protect against hemorrhagic stroke caused by brain bleeding that affects its function (Ahmad et al, 2017). Moreover, DP1 genetic ablation mitigated disease symptoms developed by a mouse model of amyotrophic lateral sclerosis (ALS) (de Boer et al, 2014). DP1 inhibition mitigates the increase in activated/amoeboid microglia associated with both hemorrhagic stroke and ALS (de Boer et al, 2014; Ahmad et al, 2017). In conclusion, modulating DP1 function is a promising therapeutic strategy applicable to different types of brain conditions and injuries related to AD.

In our studies with 11-mo Tg-AD and WT rats, we confirmed that the DP2 receptor is highly expressed in hippocampal neurons, as previously shown by others (Liang et al, 2005). DP2 is also expressed in astrocytes (Mohri et al, 2006), but was not detected in microglia (Cohen et al, 2013). Because astrocyte levels were stable in the hippocampus of Tg-AD compared with WT rats, we focused our studies on DP2 and neuronal levels. Both neuronal and DP2 levels decreased significantly in a parallel manner in the GCL of the hippocampal DG region of Tg-AD rats compared with controls. We propose that the decline in DP2/neuronal levels is tied to the rise in activated/amoeboid microglia. Thus, chronic PGD2 release as a result on enhanced neuroinflammation linked to AD could on the one hand damage neurons via its DP2 receptor, and on the other hand increase the levels of activated/amoeboid microglia via its DP1 receptor. The changes in DP1 and DP2 levels that we report here are regional and specific, as they were only detected in the hilar subregion and GCL of the hippocampal DG area (Fig 11).

RNAseq analysis of 33 genes involved in the PGD2 and PGE2 pathways, demonstrated that mRNA transcript levels in whole (ventral and dorsal combined) hippocampal tissue were the highest for L-PGDS. Expression of L-PGDS is up-regulated in AD phenotypes, correlates with A$\beta$ plaque burden, and is associated with pathological traits of AD, but not with ALS or Parkinson's disease (Kanekiyo et al, 2007; Kannaian et al, 2019). In our current studies, L-PGDS mRNA and protein levels were similar in WT and Tg-AD rats. However, whether changes occur in individual cell types and/or in specific hippocampal regions, like for the PGD2 receptors, remains to be determined.

L-PGDS also known as $\beta$-trace, is the primary PGD2 synthase in the brain, and is one of the most abundant (26 $\mu$g/ml, 3% of total) CSF proteins, second only to albumin (Kannaian et al, 2019; Urade, 2021). L-PGDS has a dual function, as it produces PGD2 and also acts as a lipophilic ligand-binding protein (Urade, 2021). L-PGDS is a major endogenous A$\beta$ chaperone that inhibits A$\beta$40/42 aggregation in vitro and in vivo, the latter when administered to mice intraventricularly infused with A$\beta$42 (Kanekiyo et al, 2007). In vitro studies also demonstrated that L-PGDS acts as a disaggregase by disassembling A$\beta$ fibrils (Kannaian et al, 2019). In the PNS, L-PGDS contributes to myelination during development (Trimarco et al, 2014) and potentially acts as an anti-inflammatory agent under conditions of peripheral nerve injury (Forese et al, 2020). In the latter studies, L-PGDS modulated the expression of the transcription factor Sox-2, which in the CNS regulates oligodendrocyte proliferation and differentiation (Zhang et al, 2018), and in the PNS is a negative regulator of myelination (Florio et al, 2018; Forese et al, 2020). Of the 33 PG-associated genes that we focused on in our RNAseq analysis, Sox-2 expression was the third highest after L-PGDS and prostaglandin E synthase 3. Tg-AD males exhibited a decline in Sox-2 levels which suggest that there may be impaired neurogenesis (Sarlak & Vincent, 2016). Other than neurogenesis, Sox-2 is proposed to act as a protective factor in AD, as (1) it interacts with APP and mediates $\alpha$-secretase activation in human cells, (2) its down-regulation in adult mouse brains induces neurodegeneration, and (3) its expression is down-regulated in the brains of AD patients (Sarlak et al, 2016; Sarlak & Vincent, 2016)." The expression of Sox-2 was the only one that was significantly different between male Tg-AD and WT male rats. Overall, more research is needed to establish whether modulating L-PDGS and Sox-2 has potential for preventing or treating AD.

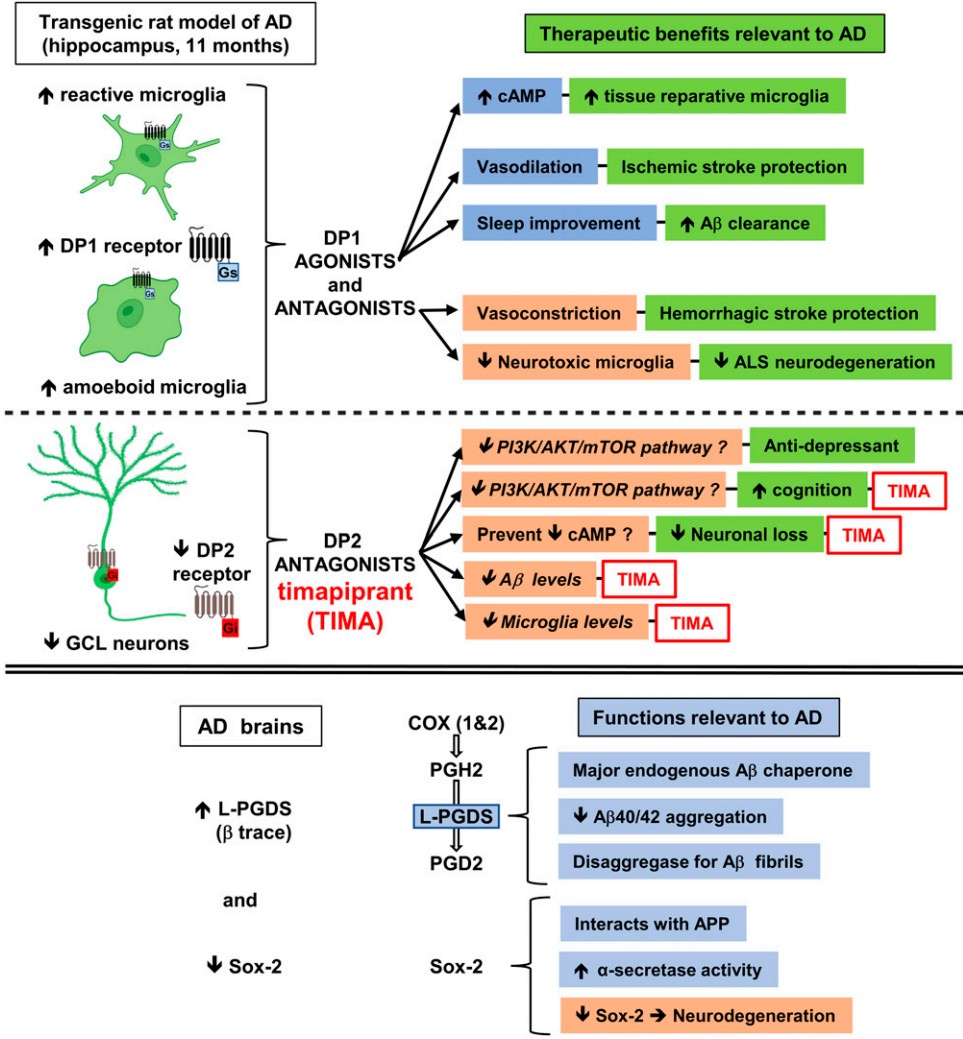

**Figure 11. Scheme depicting the relevance of the PGD2 pathway in AD and its potential as an AD therapeutic target demonstrated by timapiprant (TIMA), a DP2 antagonist.** Tg-AD rats at 11 mo of age exhibit AD pathology. We propose that manipulating PGD2 signaling through, for example, DP2 receptor antagonists such as TIMA, could prevent/mitigate AD pathology by several mechanisms. Further details are presented in the discussion, based on our current results with the Tg-AD rat model treated with TIMA, and studies published by others. Figure partially created with BioRender.com. Source data are available for this figure.

Our results from treating Tg-AD rats with timapiprant reveal that manipulating PGD2 signaling with DP2 antagonists could potentially mitigate plaque load, neuronal loss and microgliosis, in addition to improving cognitive outcomes in AD patients (Fig 11). In the brain, DP2 activation accelerates damage, as corroborated by studies with rat hippocampal neuronal and organotypic cultures in paradigms of glutamate toxicity (Liang et al, 2005, 2007) or aluminum overload (Ma et al, 2016), and in a rat model of type 2 diabetes (Yang et al, 2021). In the latter study, DP2 signaling promoted brain damage and inhibited autophagy by activating the PI3K/AKT/mTOR pathway (Yang et al, 2021). Moreover, DP2 signaling mediates depression as well as cognitive dysfunction, supported by DP2-deficient mice exhibiting anti-depressant–like activity in a chronic corticosterone-induced model of depression (Onaka et al, 2015) and improved cognition in an NMDA receptor antagonist-induced model of cognitive dysfunction (Onaka et al, 2016). These findings support that DP2 signaling has a negative impact on emotion and cognition. Thus, selective DP2 receptor antagonists may represent an encouraging option for treating some types of brain disorders.

However, further studies are necessary to establish the long-term safety and benefits of these drugs that could be used as a monotherapy or in combination with other therapies aimed, for example, at reducing amyloid plaque burden in AD.

In conclusion, at the periphery, PGD2 is an established inflammatory mediator (Mohri et al, 2007), and its effects include enhancing vascular permeability (Flower et al, 1976), modulating chemotaxis (Hirai et al, 2001), antigen presentation (Hervé et al, 2003), vasodilatation, bronchoconstriction, platelet aggregation, glycogenolysis, allergic reaction, and intraocular pressure (Ahmad et al, 2019), as well as resolving peripheral nerve injury (Forese et al, 2020). In the CNS, PGD2 regulates sleep induction, body temperature, olfactory function, nociception, neuromodulation, and protects the brain from ischemic stroke (Ahmad et al, 2019). Our current data suggest that, as an alternative to NSAIDs and as a novel approach for treating neuroinflammation, manipulating PGD2 signaling with, for example, DP2 receptor antagonists, could have a significant translational and multifactorial potential as a therapeutic for AD.

# Materials and Methods

### TgF344-AD transgenic rat model of AD

Fisher transgenic F344-AD (Tg-AD) rats (Cohen et al, 2013) express human Swedish amyloid precursor protein (APPswe) and Δ exon 9 presenilin-1 (PS1ΔE9) driven by the prion promoter, at 2.6- and 6.2-fold higher levels, respectively, than the endogenous rat proteins (Cohen et al, 2013). We purchased the Tg-AD rats and their WT littermates from Rat Resource and Research Center (RRRC) at 4 wk of age. The rats were housed in pairs upon arrival and maintained on a 12 h light/dark cycle with food and water available ad libitum. The Institutional Animal Care and Use Committee at Hunter College approved all animal procedures.

The Tg-AD rats exhibit a progressive age-dependent AD-like pathology as depicted in Fig 12 and described in Cohen et al (2013), including cognitive deficits, neuronal loss, Aβ plaque, and neurofibrillary tangle burden, as well as gliosis. No differences in pathology were reported between sexes (Cohen et al, 2013).

### Experimental design

A total of 93 rats for the combined female and male studies (WT n = 49 [27 females, 22 males], Tg-AD n = 44 [25 females, 19 males]) across multiple cohorts were used (Fig 12). For the timapiprant-treated studies, 9 WT and 9 Tg-AD males were used. At 7 mo, Tg-AD and WT rats began timapiprant treatment (Cat. no. HY-15342; MCE) with 15 mg/kg body weight/day/rat administered orally in rodent chow (Research Diets Inc.) for 4 mo, thus rats were euthanized at 11 mo of age. Dosage curves for WTTR compared with TGTR rats and weight changes during treatment are shown in Figs S12 and S13, respectively. Future studies will include timapiprant-treated females.

We evaluated all rats at 11 mo of age as described in Fig 12. Hippocampal-dependent cognitive deficits were estimated with the radial 8-arm maze (RAM), which is a passive behavioral task, and/or the active-place avoidance task (aPAT). After behavioral testing, the rats were euthanized, the brains rapidly isolated and bisected into hemispheres, and processed for the different assays as described below and in Fig 12.

### Tissue collection and preparation

At 11 mo of age, the rats were anesthetized with an intraperitoneal injection containing ketamine (100 mg/kg body weight) and xylazine (10 mg/kg body weight), and then transcardially perfused with chilled RNAase free PBS. The brain left hemispheres were micro-dissected into different regions, snap-frozen with a CoolRack over dry ice, and the hippocampal tissue used for mass spectrometry, RNAseq, or Western blot analyses. Whole right brain hemispheres were placed in a 4% paraformaldehyde/PBS solution for 48 h at 4°C, followed by cryo-protection with a 30% sucrose/PBS solution to prevent water-freeze damage, and then flash-frozen using 2-methylbutane, and stored at −80°C until sectioning for IHC.

### LC–MS/MS for PG quantification

Rat hippocampal tissue from 11-mo WT (n = 31) and Tg-AD (n = 32) rats were analyzed by quantitative LC–MS/MS to determine PGD2,

PGE2, PGJ2, and thromboxane B2 (TxB2) concentrations using the standard calibration curves for each compound. Samples were prepared as previously described (Avila et al, 2018). In summary, hippocampal tissues were homogenized in PBS using a BeadBug microtube homogenizer, then a 10-mg wet weight equivalent of homogenate was removed and further diluted 1:1 with 1% formic acid. Deuterated internal standards were added and loaded on a Biotage SLE+ cartridge and were eluted twice with t-butyl methyl ether. The eluent was spiked with a trap solution consisting of 10% glycerol in methanol with 0.01 mg/ml butylated hydroxytoluene. Samples were dried in a speed vacuum at 35°C, the tubes were washed with hexane and re-dried. The residue was dissolved in 80:20 water:acetonitrile with butylated hydroxytoluene and spin fil-tered with a 0.22-μm Millipore Ultrafree filter. 30 μl of sample was analyzed. Prostaglandin standard curves were spiked into PBS and prepared identically to the samples. Area ratios were plotted and unknowns determined using the slopes.

PGs were analyzed using a 5500 Q-TRAP hybrid/triple quadru-pole linear ion trap mass spectrometer (Applied Biosystems) with electrospray ionization (ESI) in negative mode as previously de-scribed (Avila et al, 2018). The mass spectrometer was interfaced to a Shimadzu SIL-20AC XR auto-sampler followed by 2 LC-20AD XR LC pumps. The scheduled MRM transitions were monitored within a 1.5 min time-window. Optimal instrument parameters were de-termined by direct infusion of each analyte. The gradient mobile phase delivered at a flow rate of 0.5 ml/min, consisted of two solvents, 0.05% acetic acid in water and acetonitrile. The analytes were resolved on a Betabasic-C18 (100 × 2 mm, 3 μm) column at 40°C using the Shimadzu column oven. Data were acquired using Analyst 1.5.1 and analyzed using Multiquant 3.0.1(AB Sciex).

### Immunohistochemistry

Coronal sections were sliced into 30 μm sections using a cryostat (Leica CM3050 S). IHC was restricted to dorsal hippocampal tissue within the following Bregma coordinates: −3.36 to −4.36 mm (Paxinos & Watson, 2007). Sections were mounted on gelatin slides and immunostained as previously described (Avila et al, 2020). After immunostaining, a mounting media of VectaShield with DAPI (# H-1200-10; Vector Labs) was used and slides were stored in the dark at 4°C until imaged. Sections were viewed on a Zeiss Axio Imager M2 with AxioVision software to capture ZVI files of 10× and 20× mosaic images of the whole hippocampus, and then converted to TIF files. Signal density (O.D.) was quantified using ImageJ as previously described (Corwin et al, 2018).

Two to three sections (averaged) from each rat were immuno-stained with either a combination of anti-DP1 and anti-Iba1 antibodies or anti-DP2 and anti-NeuN antibodies. Primary and secondary antibodies are listed in Table S10. For quantification, the following thresholds were used: *DP1*: mean + 1.5*std, particles analyzed were in the range: 10–10,000, and circularity: 0–1.00; *Iba1*: mean + 1.5*std, particles analyzed were in the range: 50–8,000, and circularity: 0–1.00; *DP2*: mean + 1.5*std, particles analyzed were in the range: 10–5,000, and circularity: 0–1.00; *NeuN*: mean + 1.5*std, particles analyzed were in the range: 10–10,000, and circularity: 0–1.00.

## Experimental Design

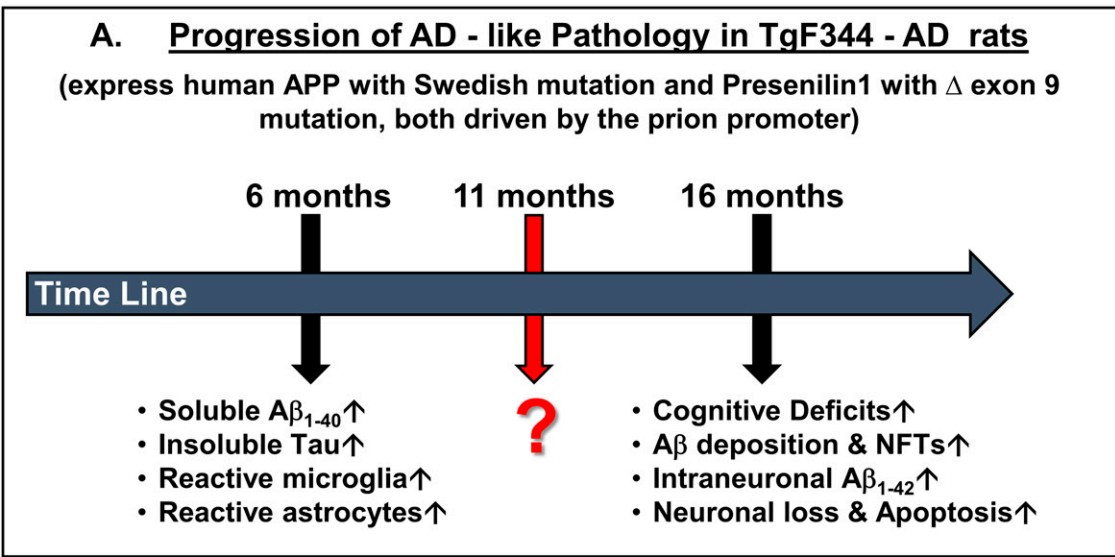

**Figure 12. Schematic representation of the experimental design.**
**(A)** Time line of the progression of the AD pathology developed by Tg-AD rats. We investigated the AD pathology of Tg-AD rats at 11 mo of age. **(B)** Rat groups used in the study. **(C)** Assessments of the AD pathology developed by the Tg-AD rats at 11 mo of age. **(D)** Timapiprant treatment overview.

Colocalization was analyzed by measuring the overlap of the masks for the two channels. For DP1 and Iba1 we report the number of microglia co-localized with DP1 per specific area ($nm^2$). For DP2 and NeuN we report the percentage DP2 and NeuN signals co-localized within a specific area ($nm^2$). In addition, Iba1+ ramified, reactive, and amoeboid microglia phenotypes were analyzed for circularity based on

the ImageJ form factor (FF = $4\pi$ × area/perimeter$^2$): ramified (FF < 0.50), reactive (FF: 0.50–0.70), and amoeboid (FF > 0.70) (Corwin et al, 2018).

Astrocytes were immunostained with an anti-GFAP antibody as listed in Table S10. For quantification, the following thresholds were used: mean + 1.5*std, particles analyzed were in the range: 30–1,000, and circularity 0–1.00.

## RNAseq analysis

Hippocampal tissue was used for RNAseq analysis outsourced to the UCLA Technology Center for Genomics & Bioinformatics services. Samples from five male WT and five male Tg-AD rats were compared, and the same for female rats. Briefly, total RNA was isolated from the hippocampal tissue using the RNeasy Mini Kit from QIAGEN. The integrity of total RNA was examined by the Agilent 4200 TapeStation System. Libraries for RNAseq were constructed with the Kapa Stranded mRNA Kit (Cat. no. KK8421; Roche) to generate strand-specific RNAseq libraries, which were amplified and sequencing was performed with the HiSeq3000 sequencer. Gene expression data were normalized as reads per million (RPM) using the TMM method. Differentially expressed genes between WT and Tg-AD rats for each sex were determined using the edgeR program (Robinson et al, 2009). RPMs were analyzed for fold-change, *P*-values, and FDR for each gene (Table S8).

## Western blot analysis

Hippocampal tissue (20–25 mg) was homogenized in TBS (with protease/phosphatase inhibitors) for 90 s at 25°C with the Bedbug Microtube Homogenizer (3,400 rpm, Model D1030; Benchmark Scientific). The supernatant was stored for 16 h at –80°C, followed by centrifugation at 14,000 rpm for 20 min at 4°C. The supernatant was filtered using biomasher homogenizer tubes (#09-A10-050; OMNI International). Samples were stored at –80°C until use. Protein concentration was determined with the BCA assay (Pierce Biotechnology), followed by normalization. Either 30 $\mu$g (for DP2, PPAR$\gamma$, L-PGDS, Sox-2, and COX-2) or 50 $\mu$g of protein (for DP1) from each sample were run on 4–12% SDS gels and transferred to nitrocellulose membranes with the iBlot dry blotting system (Life Technologies) for 7 min. Membranes were blocked with SuperBlock (#37535; Thermo Fisher Scientific), and hybridized with various primary antibodies followed by HRP-conjugated secondary antibodies (Table S10), before developing with an enhanced chemiluminescence (ECL) substrate (SuperSignal West Pico PLUS, #34580; Thermo Fisher Scientific), and detected on a BX810 autoradiography film (Midwest Scientific). ImageJ software (Rasband, W.S., ImageJ, U. S. National Institutes of Health, https://imagej.nih.gov/ij/, 1997–2018) was used for semi-quantification by densitometry of the respective bands. Loading controls used were GAPDH, tubulin, or $\beta$-actin depending on their molecular weights to avoid overlapping with the other proteins studied.

## Cognitive behavior assessment with the passive radial 8-arm maze

This variant of RAM is a passive task that uses positive reinforcement (food) to assess spatial working memory. This RAM is classified as working memory because only short-term memory is used and memory of previous trial baits will not aid the rat in later trials as all baits are used and replenished after each trial. This hippocampal dependent task uses spatial cues in the test room. The maze is divided into eight arms with a bait of food (Ensure Food Supplement) at the end of each arm in a submerged food cup. Before training, rats were food deprived to 85% of their ad libitum

body weight and received six shaping trials across 2 d. For training, the rats were tested four times across 2 d. The rats begun the training confined to the center of the arena with an opaque covering. Once the opaque covering was removed, the rat was free to start the trial to collect all eight baits. Entrances were recorded after the rat crossed halfway across the arm towards the bait. When the rat returned to a bait that was previously consumed this was deemed as an error. Animals were required to collect all eight baits for the trial to end, and if the trial exceeded 25 min, the trial was not included in the analysis. After each trial, the maze was shifted at 90° and cleaned with a 70% ethanol solution to prevent internal maze cues being used. To prevent the rats using their sense of smell to find baits, the maze room had ample food placed throughout the maze room. Data of all fully completed training trials were analyzed.

## Cognitive behavior assessment with the active place avoidance task

This variant of aPAT is an active task that uses negative reinforcement (shock) to access spatial learning. This aPAT is classified as reference memory because long-term spatial learning is used as the rats experience repeated trials with a fixed shock quadrant, so referencing previous trials will aid in better performance. The task challenges the rat to avoid a fixed quadrant of the arena as the arena rotates at one revolution per minute. A computer-controlled system was used for aPAT (Bio-Signal Group). The arena used for this task was enclosed with a transparent plastic wall that was fixed to the arena. An overhead camera (Tracker; Bio-Signal Group) was calibrated to the white hue of the rats and tracked the rat's movement. This hippocampal-dependent task used spatial cues in the test room. The rotating arena forced the rat into the fixed quadrant. After the system detected that the rat was in the fixed quadrant for 1.5 s, the system delivered a pulse shock of 0.2 mA throughout the arena every 1.5 s, giving the rat a foot shock and subsequent foot shocks until it left the fixed quadrant. The rotating arena forced the rat to actively avoid the fixed quadrant, otherwise it would receive a shock. This hippocampal-dependent task uses spatial cues to help the rat navigate within the spatial environment. Before training, rats were habituated to the rotating arena for 10 min without a shock. For training, the rats received six 10-min trials with 10-min breaks in their home cage between every trial. To access retention, on the next day the rats received a 10-min trial without a shock zone. The system software recorded data for all trials, and all data were exported to .tbl files and analyzed offline (TrackAnalysis, Bio-Signal Group).

## Statistics

All data are represented as the mean ± SEM. Statistical analyses were performed with GraphPad Prism 9 (GraphPad Software). All *P*-values, SEMs, and *t*-statistics are shown on graphs and/or in supplemental tables. Welch's unpaired one-tailed *t* test was used to compare means between the two groups (WT and Tg-AD) for PG (Fig 1), IHC (Figs 2–5 and Tables S1–S7), WB (Fig 7 and Table S9), RAM (Fig 8), and the two groups (TGNT, Tg-AD non-treated, and TGTR, Tg-AD timapiprant-treated males, Fig 10). Multiple unpaired *t* test was used for RNAseq (Fig 6 and Table S8) for the 33 PG genes with an FDR set to 1% using the two-stage step-up method (Benjamini, Krieger, and Yekutieli). Multi-factor comparisons for aPAT (Figs 9 and 10) were performed using a two-way repeated

measure ANOVA, followed by a post hoc (Sidak's) to access differences across individual training trials or conditions.

## Supplementary Information

## Acknowledgements

We thank the technical support of Osama Chaudry[a], Rushna Snetha[a], Aminoor Rashid[a] (all Master's students), and Amber Alliger[b] (Lecturer) in the Departments of Biological Sciences[a] and Psychology[b] at Hunter College, CUNY. We also thank Ms. Lisa Bleyle and Dr. Dennis Koop for the assistance with the prostaglandin analysis conducted in the Bioanalytical Shared Resource/Pharmacokinetics Core. The facility is part of the University Shared Resource Program at Oregon Health. This work was supported in part by NIH/NIA R01AG057555 to L Xie, NIH training grants 5T32HL135465-A1 to support CH Wallace and R25GM060665 to support G Oliveros, and the City University of New York (PhD program in Biochemistry, Graduate Center).

### Author Contributions

CH Wallace: conceptualization, data curation, formal analysis, investigation, methodology, and writing—original draft, review, and editing.
G Oliveros: data curation, formal analysis, investigation, and methodology.
PA Serrano: conceptualization, data curation, formal analysis, funding acquisition, methodology, and writing—original draft, review, and editing.
P Rockwell: conceptualization, data curation, formal analysis, funding acquisition, methodology, and writing—original draft, review, and editing.
L Xie: resources, software, and funding acquisition.
M Figueiredo-Pereira: conceptualization, resources, data curation, formal analysis, funding acquisition, validation, investigation, methodology, and writing—original draft, review, and editing.

### Conflict of Interest Statement

The authors declare that they have no conflict of interest.

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
