## [Reviewer comments · Life Science Alliance]

Life Science Alliance

Timapiprant, a prostaglandin D2 receptor antagonist, ameliorates pathology in a rat Alzheimer's model

Charles Wallace, Giovanni Oliveros, Peter Serrano, Patricia Rockwell, Lei Xie, and Maria Figueiredo-Pereira

DOI: <https://doi.org/10.26508/lsa.202201555>

Corresponding author(s): *Maria Figueiredo-Pereira, Hunter College*

Review Timeline:

Submission Date:	2022-06-08
Editorial Decision:	2022-07-21
Revision Received:	2022-09-04
Editorial Decision:	2022-09-06
Revision Received:	2022-09-12
Accepted:	2022-09-13

Transaction Report:

July 21, 2022

Re: Life Science Alliance manuscript #LSA-2022-01555-T

Maria E. Figueiredo-Pereira
Hunter College
Biological Sciences
695 Park Avenue
Room 827N
New York, NY 10065

Dear Dr. Figueiredo-Pereira,

Thank you for submitting your manuscript entitled "Prostaglandin DP2 receptor antagonist ameliorates pathology in a rat model of Alzheimer's disease" to Life Science Alliance. The manuscript was assessed by expert reviewers, whose comments are appended to this letter. We invite you to submit a revised manuscript addressing the Reviewer comments.

Thank you for this interesting contribution to Life Science Alliance. We are looking forward to receiving your revised manuscript.

Sincerely,

B. MANUSCRIPT ORGANIZATION AND FORMATTING:

Reviewer #1 (Comments to the Authors (Required)):

In this paper, the authors conducted two general types of experiments. The key experiment was to test effects of timapiprant on behavioral endpoints, neuropathology (neuronal loss, accumulation of A β , microglial activation), in a rat model of Alzheimer's disease (AD). The key conclusion of the study is that a prostaglandin D2 receptor antagonist improved behavioral and histologic outcomes in the rat AD model. The data are reasonably convincing that the inhibitor reduced A β , increased NeuN staining and reduced microglia. Effects of the inhibitor on behavioral outcomes are less convincing. If the ANOVA showed a main effect for drug confidence in the significance of the improved latency to first entrance would be improved.

The remainder of the paper contains a systematic and comprehensive characterization of changes in prostanoid levels and expression of prostanoid receptors is also presented. For the most part, these studies recapitulate prior work in other brain regions or model systems. The way the data on mRNA expression for the 21 factors involved in prostanoid formation and inactivation was presented made it difficult to appreciate differences between control and AD rats. The supplemental table is dense but does seem to have the requisite information. I commend the authors on the careful way in which the studies were done.

The manuscript would be improved if the following minor comments regarding presentation and organization of data were addressed.

1. The figure legends contain varying degrees of description of the results. They should be revised to describe what is shown in the figure, and in each panel.
2. Where ANOVA is used the figure legends would be improved by showing main effects and interactions either as an insert in the figure or by listing these data in the figure legend. Main effects should be mentioned in Results as evidence supporting an effect of the medication.
3. The presentation of the RNAseq data as pie charts is a little confusing because it is not clear if reads are shown for each sector of the pie.

Reviewer #2 (Comments to the Authors (Required)):

1. This study by Wallace and colleagues is a concise investigation of the prostaglandin D2 (PGD2) signaling pathway in the context of neuroinflammation in AD. This study provides the field with updated information on the role of PDG2 in the context of neuroinflammation in AD and the data suggest that as an alternative to NSAIDs and as a novel approach for treating neuroinflammation, manipulating PGD2 signaling with DP2 receptor antagonists, could have a significant translational and multifactorial potential as a therapeutic for AD. This provides the field with a foundation to further explore the role of PGD2 and its associated receptors/molecular pathways and its contribution to inflammation that underlies neurodegenerative disease processes in AD.

2. All of the main points of the paper are supported by the data, however it is important to understand why the authors did not use a panel of microglial markers for microglial analysis, having only used IBA1 which is a pan-microglial marker, Given the increasing understanding of various subtypes and spectrum of microglial activation states particularly in the context of neuroinflammation in neurodegenerative diseases such as AD, use of other more specific microglial markers for analysis of microglial morphology (ramified, amoeboid etc) and phenotype (residential, activated etc.) would have been beneficial to strengthen the results pertaining to microglial assessment using immunohistochemistry. If the authors can address this limitation adequately in text then further immunostaining/experiments are not necessary. Examples of other more specific markers that could have been used include tmem119, Sall1, CD68 etc.

3. Overall the article is well written, with sound use of figures and tables to support data. However, supplementary Fig. 9 'Monoclonal 6E-10n' western blots are not good quality for publication and appear to be overexposed to show bands. A higher quality blot would be of benefit, however this is supplementary data so would be up to the editors discretion.

Reviewer #3 (Comments to the Authors (Required)):

This study investigated the relevance of the prostaglandin D2 (PGD2) signalling pathway in Alzheimer's disease (AD), using the transgenic rat TgF344-AD (Tg-AD) as model. These rats exhibit age-dependent and progressive AD pathology, including neuroinflammation, tau tangles and neuronal loss (Cohen et al., 2013).

The authors investigated the relative levels of the four prostaglandins: PGD2, PGE2, PGJ2 and thromboxane B2 as well as the cellular distribution of the linked prostaglandin DP1 and DP2 receptors. Their studies confirm that PGD2 is the most abundant prostaglandin in the brain and reveal significant differences in DP1 vs DP2 receptor levels in microglia and neurons, respectively, between Tg-AD and control rats.

They also assessed the mRNA levels of 33 genes involved in the PGD2 and PGE2 pathways in hippocampal tissue from 11-month-old WT and Tg-AD rats. The transcriptome analysis identified the lipocalin-type prostaglandin D synthase (L-PGDS, the primary synthase for PGD2) as the most abundant mRNA in the rat hippocampus, independently of genotype. Comparison between WT and Tg-AD animals showed no significant differences in mRNA expression levels for most of the tested genes excepting Sox-2, which was significantly downregulated in male Tg-AD rats compared to their WT littermates.

In addition, the authors investigated amyloid- β (A β) plaque burden, neuronal loss, microgliosis and cognitive performance in the transgenic animals. Their findings confirmed previous analysis (Cohen et al., 2013), and revealed that at 11 months of age the Tg-AD rats exhibit microglial activation, neuronal loss, early tau pathology (tau-PHF) and impaired hippocampal-dependent spatial learning and memory in two independent tests. Most importantly, neuronal loss specifically affected discrete hippocampal regions (DG, GCL and CA3c subregions) and Tg-AD rats presented more of the reactive and amoeboid microglial types, compared with WT controls, in the hippocampus (GD). Furthermore, the transgenic rats had higher DP1 co-localization with ramified, reactive and amoeboid microglia, respectively; while astrocytic levels were similar. The levels of DP2 were higher in neurons of the hippocampal CA1 region of Tg-AD rats, but decreased in the granular cell layer (GCL), compared to WT controls.

Notably, the DP2 antagonist timapiprant significantly ameliorated A β plaque load in the hippocampal DG subregion in treated vs nontreated transgenic rats. Timapiprant treatment also alleviated neuronal loss in the GCL subregion and microgliosis in the hilar subregion of the transgenic Tg-AD rats. Based on these findings, the authors concluded that timapiprant could have a significant translational and multifactorial potential as a therapeutic for AD.

Major points:

1. The main concern of this reviewer is that the manuscript is difficult to read.

The study presents extensive analyses related to the characterization of the TgF344-AD model, the analysis of the prostaglandin D2 (PGD2) signalling pathway in the brain of rats and the effects of the DP2 antagonist timapiprant on AD pathology. However, the authors only introduce the studies related to PGD2 signalling and the use of antagonists timapiprant. The relatively extensive characterization of the TgF344-AD model is presented but not motivated. This reviewer suggests to revise the introduction in order to explain better the rationale of the presented analyses.

2. To improve readability this reviewer suggests to revise the following points:

-In the title, write 'Prostaglandin D2 receptor' in full in the title, instead of DP2;

-The abstract contains too many abbreviations, it would be better to write full-names;

-The subsection "Tg-AD rats display neuronal and DP2 receptor loss in the hippocampus" is not clear and needs revision.

This reviewer suggests to merge the results and discussion sections in order to improve readability and clarity. If this is not possible, at least explain the implications of the significant differences in mRNA expression levels of Sox-2 in the results section.

Reply to Reviewers: We thank the reviewers for the very helpful comments that contributed to a significantly improved manuscript. We revised the manuscript as suggested by the reviewers and address below each of the reviewers' concerns.

Reviewer #1 (Comments to the Authors (Required):

In this paper, the authors conducted two general types of experiments. The key experiment was to test effects of timapiprant on behavioral endpoints, neuropathology (neuronal loss, accumulation of A β , microglial activation), in a rat model of Alzheimer's disease (AD). The key conclusion of the study is that a prostaglandin D2 receptor antagonist improved behavioral and histologic outcomes in the rat AD model. The data are reasonably convincing that the inhibitor reduced A β , increased NeuN staining and reduced microglia. Effects of the inhibitor on behavioral outcomes are less convincing. If the ANOVA showed a main effect for drug confidence in the significance of the improved latency to first entrance would be improved.

REPLY: The measure latency to first entrance was significant but perhaps the abbreviations made it unclear. On figure 8A we revised the main effects to not be abbreviated to help with clarity. The main effects in Figure 7 are also revised to be not abbreviated for consistency in the paper.

The remainder of the paper contains a systematic and comprehensive characterization of changes in prostanoid levels and expression of prostanoid receptors is also presented. For the most part, these studies recapitulate prior work in other brain regions or model systems. The way the data on mRNA expression for the 21 factors involved in prostanoid formation and inactivation was presented made it difficult to appreciate differences between control and AD rats.

REPLY: To address the reviewer's concern we replaced the pie graphs with bar graphs. The mRNA expression between control and AD rats for those factors was primarily not different except for example Sox-2 in males which was only significantly different between Tg-AD and WT male rats. This information is included in the first paragraph of page 17.

The supplemental table is dense but does seem to have the requisite information. I commend the authors on the careful way in which the studies were done.

REPLY: We thank the reviewer for the positive comments.

The manuscript would be improved if the following minor comments regarding presentation and organization of data were addressed.

1. The figure legends contain varying degrees of description of the results. They should be revised to describe what is shown in the figure, and in each panel.

REPLY: The statistics displayed in the figure legends match the results and more detail was added in cases where there were varying descriptions. The revised figure legends have more statistics added to them that mirror the results section. We made changes in all figure legends except Fig 1 and 9.

2. Where ANOVA is used the figure legends would be improved by showing main effects and interactions either as an insert in the figure or by listing these data in the figure legend. Main effects should be mentioned in Results as evidence supporting an effect of the medication.

REPLY: As explained above, the measure latency to first entrance was significant but perhaps the abbreviations made it unclear. On figure 8A we revised the main effects to not be abbreviated to help with clarity. The main effects in Figure 7 are also revised to be not abbreviated for consistency in the paper.

3. The presentation of the RNAseq data as pie charts is a little confusing because it is not clear if reads are shown for each sector of the pie.

REPLY: We replaced the pie graphs with bar graphs. The new Fig. 5A shows the most abundant genes and the new Fig. 5B shows the least abundant genes by reads per million (RPM) for both figures. Fig. 5A and 5B represent RNAseq results for 11-month WT females only. The pie graphs in supplementary Figures 2-5 were also replaced by bar graphs following the same criteria. These figures represent RNAseq results for 11-month WT and Tg-AD females and for 11-month WT and Tg-AD males.

Reviewer #2 (Comments to the Authors (Required):

1. This study by Wallace and colleagues is a concise investigation of the prostaglandin D2 (PGD2) signaling pathway in the context of neuroinflammation in AD. This study provides the field with updated information on the role of PDG2 in the context of neuroinflammation in AD and the data suggest that as an alternative to NSAIDs and as a novel approach for treating neuroinflammation, manipulating PGD2 signaling with DP2 receptor antagonists, could have a significant translational and multifactorial potential as a therapeutic for AD. This provides the field with a foundation to further explore the role of PGD2 and its associated receptors/molecular pathways and its contribution to inflammation that underlies neurodegenerative disease processes in AD.

REPLY: We thank the reviewer for the positive comments.

2. All of the main points of the paper are supported by the data, however it is important to understand why the authors did not use a panel of microglial markers for microglial analysis, having only used IBA1 which is a pan-microglial marker. Given the increasing understanding of various subtypes and spectrum of microglial activation states particularly in the context of neuroinflammation in neurodegenerative diseases such as AD, use of other more specific microglial markers for analysis of microglial morphology (ramified, amoeboid etc) and phenotype (residential, activated etc.) would have been beneficial to strengthen the results pertaining to microglial assessment using immunohistochemistry. If the authors can address this limitation adequately in text then further immunostaining/experiments are not necessary. Examples of other more specific markers that could have been used include tmem119, Sall1, CD68 etc.

REPLY: We agree with the reviewer that a panel of microglia markers should be used in future studies. However, there are numerous publications that support using Iba1 with fractural analysis to distinguish microglia shapes, which directly relate to their function. On page 14 of the discussion (end of second paragraph) we added two sentences that support this analysis.

“It is well agreed that microglia form factor is directly related to its function. The shift away from a highly branched ramified state is indicative of microglia changing in response to pathological conditions (Karperien, Ahammer, & Jelinek, 2013)”.

3. Overall the article is well written, with sound use of figures and tables to support data. However, supplementary Fig. 9 'Monoclonal 6E-10n' western blots are not good quality for publication and appear to be overexposed to show bands. A higher quality blot would be of benefit, however this is supplementary data so would be up to the editors discretion.

REPLY: We thank the reviewer for the positive comments. We agree with the reviewer and we fixed the 6E10 blot with a lighter exposure that was cleaner. Please see the new supplementary Fig. 9

Reviewer #3 (Comments to the Authors (Required):

This study investigated the relevance of the prostaglandin D2 (PGD2) signalling pathway in Alzheimer's disease (AD), using the transgenic rat TgF344-AD (Tg-AD) as model. These rats exhibit age-dependent and progressive AD pathology, including neuroinflammation, tau tangles and neuronal loss (Cohen et al., 2013).

The authors investigated the relative levels of the four prostaglandins: PGD2, PGE2, PGJ2 and thromboxane B2 as well as the cellular distribution of the linked prostaglandin DP1 and DP2 receptors. Their studies confirm that PGD2 is the most abundant prostaglandin in the brain and reveal significant differences in DP1 vs DP2 receptor levels in microglia and neurons, respectively, between Tg-AD and control rats.

They also assessed the mRNA levels of 33 genes involved in the PGD2 and PGE2 pathways in hippocampal tissue from 11-month-old WT and Tg-AD rats. The transcriptome analysis identified the lipocalin-type prostaglandin D synthase (L-PGDS, the primary synthase for PGD2) as the most abundant mRNA in the rat hippocampus, independently of genotype. Comparison between WT and Tg-AD animals showed no significant differences in mRNA expression levels for most of the tested genes excepting Sox-2, which was significantly downregulated in male Tg-AD rats compared to their WT littermates.

In addition, the authors investigated amyloid- β (A β) plaque burden, neuronal loss, microgliosis and cognitive performance in the transgenic animals. Their findings confirmed previous analysis (Cohen et al., 2013), and revealed that at 11 months of age the Tg-AD rats exhibit microglial activation, neuronal loss, early tau pathology (tau-PHF) and impaired hippocampal-dependent spatial learning and memory in two independent tests. Most importantly, neuronal loss specifically affected discrete hippocampal regions (DG, GCL and CA3c subregions) and Tg-AD rats presented more of the reactive and amoeboid microglial types, compared with WT controls, in the hippocampus (GD). Furthermore, the transgenic rats had higher DP1 co-localization with ramified, reactive and amoeboid microglia, respectively; while astrocytic levels were similar. The levels of DP2 were higher in neurons of the hippocampal CA1 region of Tg-AD rats, but decreased in the granular cell layer (GCL), compared to WT controls.

Notably, the DP2 antagonist timapiprant significantly ameliorated A β plaque load in the hippocampal DG subregion in treated vs nontreated transgenic rats. Timapiprant treatment also alleviated neuronal loss in the GCL subregion and microgliosis in the hilar subregion of the transgenic Tg-AD rats. Based on these findings, the authors concluded that timapiprant could have a significant translational and multifactorial potential as a therapeutic for AD.

Major points:

1. The main concern of this reviewer is that the manuscript is difficult to read.

The study presents extensive analyses related to the characterization of the TgF344-AD model, the analysis of the prostaglandin D2 (PGD2) signaling pathway in the brain of rats and the effects of the DP2 antagonist timapiprant on AD pathology. However, the authors only introduce the studies related to PGD2 signaling and the use of antagonists timapiprant. The relatively extensive characterization of the TgF344-AD model is presented but not motivated. This reviewer suggests revising the introduction in order to explain better the rationale of the presented analyses.

REPLY: We revised the introduction on page 4 to give a better rationale for why these studies were necessary. We wanted to investigate in an AD setting if overall PGD2 levels were changing as well as the receptors that PGD2 binds to. We added the following sentence to the second paragraph on page 4: Since the literature on PGD2 and its relevance to AD is limited, we investigated the importance of the PGD2 pathway in AD with the TgF344-AD (Tg-AD) rat model that closely mirrors AD in humans, specifically where neuronal loss and gliosis are detected.

2. To improve readability this reviewer suggests to revise the following points:

-In the title, write 'Prostaglandin D2 receptor' in full in the title, instead of DP2;

REPLY: The total length of the title should not exceed 100 characters (including spaces). According to the reviewer's suggestion we revised the title to "Timapiprant a prostaglandin D2 receptor antagonist ameliorates pathology in a rat Alzheimer's model"

-The abstract contains too many abbreviations, it would be better to write full-names;

REPLY: The abstract cannot exceed 175 words. According to the reviewer's suggestion we removed most of the abbreviations.

-The subsection "Tg-AD rats display neuronal and DP2 receptor loss in the hippocampus" is not clear and needs revision.

REPLY: The subsection was revised to be more precise with language. Please see:

Title (last paragraph on page 7): Tg-AD rats display a loss of neurons as well as lower DP2 levels in the hippocampal granular cell layer

First paragraph on page 8: It is clear that at least 50% of DP2 is co-localized with neurons, as shown in Fig. 4A (yellow). However, co-localization of the DP2 receptor with neurons was not significantly different between WT and Tg-AD rats in any hippocampal regions. This further supports that NeuN and DP2 are co-localized, because both markers decrease in the GCL in tandem in Tg-AD compared to WT rats (Supplemental Table 7)."

This reviewer suggests to merge the results and discussion sections in order to improve readability and clarity. If this is not possible, at least explain the implications of the significant differences in mRNA expression levels of Sox-2 in the results section.

REPLY: Unfortunately merging the results and discussion would lead to major changes. However, the reviewer's recommendation to explain implications in Sox-2 changes was addressed in the discussion (first paragraph of page 17).

"Tg-AD males exhibited a decline in Sox-2 levels which suggests that there may be impaired neurogenesis (Sarlak & Vincent, 2016). Other than neurogenesis, Sox-2 is proposed to act as a protective factor in AD, as (1) it interacts with APP and mediates α -secretase activation in human cells, (2) its down-regulation in adult mouse brains induces neurodegeneration, and (3) its expression is downregulated in the brains of AD patients (Sarlak et al., 2016; Sarlak & Vincent, 2016)."

References

- Karperien, A., Ahammer, H., & Jelinek, H. F. (2013). Quantitating the subtleties of microglial morphology with fractal analysis. *Front Cell Neurosci*, 7, 3. Retrieved from <http://www.ncbi.nlm.nih.gov/pubmed/23386810>
- Sarlak, G., Htoo, H. H., Hernandez, J. F., Iizasa, H., Checler, F., Konietzko, U., . . . Vincent, B. (2016). Sox2 functionally interacts with β APP, the β APP intracellular domain and ADAM10 at a transcriptional level in human cells. *Neuroscience*, 312, 153-164. doi:10.1016/j.neuroscience.2015.11.022
- Sarlak, G., & Vincent, B. (2016). The Roles of the Stem Cell-Controlling Sox2 Transcription Factor: from Neuroectoderm Development to Alzheimer's Disease? *Mol Neurobiol*, 53(3), 1679-1698. doi:10.1007/s12035-015-9123-4

September 6, 2022

RE: Life Science Alliance Manuscript #LSA-2022-01555-TR

Prof. Maria E. Figueiredo-Pereira
Hunter College
Biological Sciences
695 Park Avenue
Room 827N
New York, NY 10065

Dear Dr. Figueiredo-Pereira,

Thank you for submitting your revised manuscript entitled "Timapiprant a prostaglandin D2 receptor antagonist ameliorates pathology in a rat Alzheimer model". We would be happy to publish your paper in Life Science Alliance pending final revisions necessary to meet our formatting guidelines.

- please add these commas to the title as shown: Timapiprant, a prostaglandin D2 receptor antagonist, ameliorates pathology in a rat Alzheimer's model
- please upload your main figures as single files; these will be displayed in-line in the HTML version of your paper, so please provide them as single page files (Figures 3, 4, 7 and 8 currently span more than one page); we do not have a limit on the number of main figures and these can be split if necessary for space
- please upload your supplementary figures as single files-it is fine if the supplementary figures span more than 1 page
- please use the [10 author names, et al.] format in your references (i.e. limit the author names to the first 10)
- please add the supplementary figure legends and table legends to the main manuscript text
- please add a callout for Figure S12 and Figure S13 to the main manuscript text
- please double-check your callouts for Figure 5; you have a callout for Figure 5C, but this is not in the legend or the figure itself

A. FINAL FILES:

B. MANUSCRIPT ORGANIZATION AND FORMATTING:

Sincerely,

September 13, 2022

RE: Life Science Alliance Manuscript #LSA-2022-01555-TRR

Prof. Maria E. Figueiredo-Pereira
Hunter College
Biological Sciences
695 Park Avenue
Room 827N
New York, NY 10065

Dear Dr. Figueiredo-Pereira,

Thank you for submitting your Research Article entitled "Timapiprant, a prostaglandin D2 receptor antagonist, ameliorates pathology in a rat Alzheimer's model". It is a pleasure to let you know that your manuscript is now accepted for publication in Life Science Alliance. Congratulations on this interesting work.

DISTRIBUTION OF MATERIALS:

Again, congratulations on a very nice paper. I hope you found the review process to be constructive and are pleased with how the manuscript was handled editorially. We look forward to future exciting submissions from your lab.

Sincerely,
